# ⬇STUDENTSPLAT :YOUR STUDENT MODEL LEARNS SINGLE-VIEW 3D GAUSSIAN SPLATTING

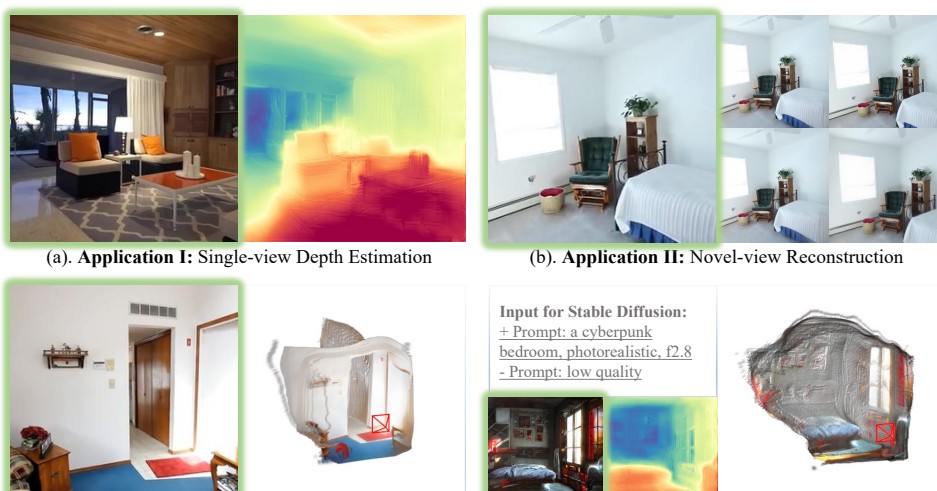

(a). **Application I:** Single-view Depth Estimation

(b). **Application II:** Novel-view Reconstruction

(c). **Application III:** 3D Gaussian Splatting

(d). **Application IV:** Text-to-3D

Figure 1: What can you do with **studentSplat**? All the results are generated using our studentSplat with teacher refine (detailed in the Appendix D) with only one input image. The input to our studentSplat is highlighted in green. studentSplat directly takes the generated image from Stable Diffusion (Rombach et al., 2022) in text-to-3D application.

## ABSTRACT

Recent advance in feed-forward 3D Gaussian splatting has enable remarkable multi-view 3D scene reconstruction or single-view 3D object reconstruction but single-view 3D scene reconstruction remain under-explored due to inherited ambiguity in single-view. We present **studentSplat**, the first single-view 3D Gaussian splatting method for scene reconstruction. To overcome the scale ambiguity and extrapolation problems inherent in novel-view supervision from a single input, we introduce two techniques: 1) a teacher-student architecture where a multi-view teacher model provides geometric supervision to the single-view student during training, addressing scale ambiguity and encourage geometric validity; and 2) an extrapolation network that completes missing scene context, enabling high-quality extrapolation. Extensive experiments show studentSplat achieves state-of-the-art single-view novel-view reconstruction quality and comparable performance to multi-view methods at the scene level. Furthermore, studentSplat demonstrates competitive performance as a self-supervised single-view depth estimation method, highlighting its potential for general single-view 3D understanding tasks.

## 1 INTRODUCTION

3D reconstruction is an essential task in robotics (Yandun et al., 2020; Han et al., 2022), navigation (Davison, 2003; Kazerouni et al., 2022), virtual reality (Bruno et al., 2010), and content cre-

ation (Jun & Nichol, 2023; Tang et al., 2023a). Advances in deep learning have enabled remarkable progress in 3D reconstruction (Sitzmann et al., 2019; Mildenhall et al., 2021; Truong et al., 2023; Kerbl et al., 2023) through per-scene optimization using a large number of views. Recently, efficient feed-forward methods (Yu et al., 2021; Charatan et al., 2024) have been proposed to take a sparse set of input views and construct the 3D reconstruction, greatly improving efficiency. However, these methods require not only multi-view input but also the corresponding camera poses. Obtaining accurate camera poses usually involves a time and computation-intensive pipeline (Ullman, 1979) and a large number of camera views or additional specialized networks (Kendall et al., 2015; Yin & Shi, 2018; Peng et al., 2019), which hinders the efficiency of feed-forward sparse-view 3D reconstruction. Single-view 3D reconstruction relaxes the requirements on both multi-view input and camera poses, serving as a more generalized alternative. Due to the inherent ambiguity in single-view input, current single-view 3D reconstruction works (Yu et al., 2021; Szymanowicz et al., 2024) only operate at the object level.

In this work, we aim to expand single-view 3D object reconstruction to the scene level and propose a model capable of performing single-view 3D scene reconstruction using only multi-view supervision (i.e., no ground truth 3D annotations). In addition to the generalizability improvements from this extension, single-view 3D scene reconstruction holds the potential to perform self-supervised single-view vision tasks such as single-view depth estimation (Li & Snavely, 2018) and aid semantic segmentation (Zhang et al., 2010; Schön et al., 2023). Finally, a single-view 3D scene reconstruction model can be applied to the results from a text-to-image generation model (Rombach et al., 2022) to achieve text-to-3D scene generation without separate training.

To enable single-view 3D reconstruction, we adopt the 3D Gaussian splatter (3DGS) (Kerbl et al., 2023) representation. We identify and address two main problems in single-view 3DGS: scale ambiguity and extrapolation. We tackle these problems by proposing **studentSplat**, the first single-view 3DGS model at the scene level. Since the unknown scale can be inferred when at least two input views are provided but is impossible using one input view (Charatan et al., 2024; Chen et al., 2024), our core design is to use a multi-view teacher model to estimate the 3D structure up to a scale and supervise the single-view student model using the teacher's estimation. Moreover, unlike a multi-view model that can bound the novel views by the input view frustums, a single-view model is required to extrapolate due to occlusion and camera view changes, which can lead to distortion of the 3DGS. We propose an extrapolator to complete the missing context in renderings before computing the novel-view reconstruction loss, both performing extrapolation and minimizing distortion. Extensive experiments show that our method can achieve good 3DGS on different benchmarks. Additionally, our method has the potential to connect 3DGS to self-supervised single-view vision tasks by demonstrating comparable performance to a self-supervised single-view depth estimation method.

Our contributions are summarized as follows:

- Propose the first single-view 3D scene Gaussian splatting model that does not require relative camera poses during inference.

- Address the extrapolation problem in single-view 3D scene reconstruction, which reduces distortion and produces out-of-context regions.

- Bridge the gap between multi-view 3D Gaussian splatting and self-supervised single-view depth estimation, expanding the applications of 3D Gaussian splatting models.

## 2 RELATED WORK

### 2.1 3D REPRESENTATION

Numerous 3D representations have been proposed to accommodate different applications. Point clouds are used in many applications (Ullman, 1979; Schönberger & Frahm, 2016; Nichol et al., 2022) where the geometric shape is important. Recently, Neural Radiance Field (NeRF) (Mildenhall et al., 2021) is proposed to learn a view-based rendering function from multi-view supervision, but this learned function does not directly represent the geometric shape. 3DGS (Kerbl et al., 2023) is an efficient alternative representation similar to point clouds. Additionally, the efficient differentiable rendering implementation of 3DGS enables direct optimization of point clouds (3D Gaussians).

This representation allows us to connect novel-view reconstruction to geometric reconstruction in an end-to-end manner.

## 2.2 FEED-FORWARD MULTI-VIEW 3D RECONSTRUCTION

NeRF (Mildenhall et al., 2021) is one of the most popular representations for multi-view 3D reconstruction. PixelNeRF (Yu et al., 2021) and GRF (Trevithick & Yang, 2021) were among the earlier works that used a feed-forward network to produce radiance fields. Subsequent approaches improved rendering performance by incorporating cross-view feature matching (Chen et al., 2021; 2023; Du et al., 2023), geometric encoding (Miyato et al., 2023), or target view information (Xu et al., 2024). Different from the predefined NeRF function, SRT (Sajjadi et al., 2022) used a transformer to represent the rendering function. Another line of work closely related to ours was initiated by pixelSplat (Charatan et al., 2024), which directly predicted 3D Gaussians from multi-view images. latentSplat (Wewer et al., 2024) improved rendering performance by operating in the latent space, while MVSplat (Chen et al., 2024) incorporated cost-volume to improve both efficiency and performance. In contrast to previous approaches, our method requires only one input view, greatly improving the generalizability and versatility of the 3DGS model. Additionally, our method connects multi-view 3DGS to single-view vision tasks by learning one model that works on both tasks.

## 2.3 FEED-FORWARD SINGLE-VIEW 3D RECONSTRUCTION

Single-view 3D reconstruction usually works at the object level. Unlike their multi-view counterparts, single-view 3D reconstruction requires extrapolation. Therefore, generative methods like diffusion models (Rombach et al., 2022; Liu et al., 2023a; Tang et al., 2023a; Liu et al., 2023b) are used to complete the reconstruction. Similar to multi-view 3D reconstruction, radiance fields are popular among the rendering methods (Liu et al., 2023a; Qian et al., 2023; Xu et al., 2023; Tang et al., 2023b; Melas-Kyriazi et al., 2023; Liu et al., 2024). TARS (Duggal & Pathak, 2022) learns the deformation between 2D images and 3D objects. Recently, more approaches (Szymanowicz et al., 2024; Tang et al., 2023a) have started using 3DGS for 3D reconstruction. Many other approaches (Nichol et al., 2022; Jun & Nichol, 2023) directly supervise the network using 3D object annotations. Additionally, some methods use learns directly from ground truth 3D annotation (Yin et al., 2021; Piccinelli et al., 2024). All existing work in single-view 3D reconstruction either requires 3D supervision or only works at the object level. In contrast, our method not only works at the scene level without 3D annotations but also has the potential to aid single-view vision tasks.

## 3 OUR APPROACH

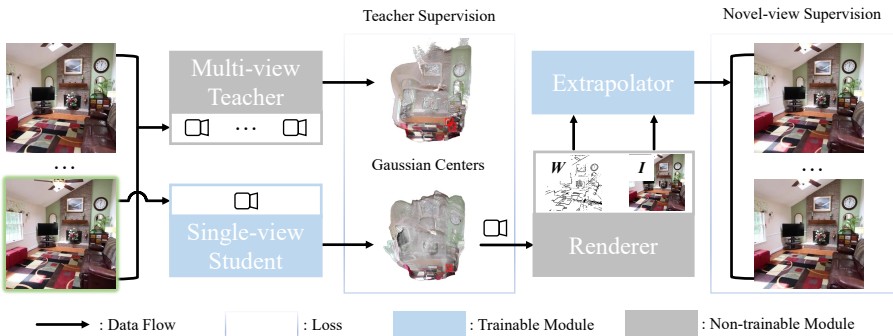

Figure 2: **The training pipeline of studentSplat.** The multi-view teacher network is used during training to produce 3D Gaussians centers (up-to an unknown scale) for geometric supervision. The input to student model is highlighted in green. The rendered student output is processed through the Extrapolator before performing novel-view supervision.

The overall pipeline is shown in Figure 2. We employ a multi-view 3DGS teacher network to provide geometric supervision, novel views to provide photometric supervision, and an extrapolator network to complete the missing context.

## 3.1 FEED-FORWARD 3D GAUSSIAN SPLATTING

In the multi-view 3DGS, we have $K$ sparse-view images $\mathcal{I} = \{\boldsymbol{I}^i\}_{i=1}^K$, ($\boldsymbol{I}^i \in \mathbb{R}^{H \times W \times 3}$) and their corresponding camera projection matrices $\mathcal{P} = \{\boldsymbol{P}^i\}_{i=1}^K$, $\boldsymbol{P}^i = \mathbf{K}^i[\mathbf{R}^i|\mathbf{t}^i]$ where $\mathbf{K}^i$, $\mathbf{R}^i$ and $\mathbf{t}^i$ are the intrinsic, rotation, and translation matrices. A multi-view 3DGS model $f_{\boldsymbol{\theta}}^K$, where $K$ is the number of views, maps images to 3D Gaussian parameters using

$$f_{\boldsymbol{\theta}}^K : \{(\boldsymbol{I}^i, \boldsymbol{P}^i)\}_{i=1}^K \mapsto \{(\boldsymbol{\mu}^j, \alpha^j, \boldsymbol{\Sigma}^j, \boldsymbol{c}^j)\}_{j=1}^{H \times W \times K}. \tag{1}$$

On the other hand, the relaxed version, the single-view 3DGS model $f_{\boldsymbol{\theta}}^1$, performs the following:

$$f_{\boldsymbol{\theta}}^1 : \boldsymbol{I}^i \mapsto \{(\boldsymbol{\mu}^j, \alpha^j, \boldsymbol{\Sigma}^j, \boldsymbol{c}^j)\}_{j=1}^{H \times W \times 1}. \tag{2}$$

Unlike the multi-view 3DGS model, the single-view 3DGS model $f_{\boldsymbol{\theta}}^1$ is more prone to scale ambiguity and extrapolation issues. To train our studentSplat, we use both geometric and photometric supervisions:

$$\mathcal{L}_{studentSplat} = \underbrace{\mathcal{L}_{geo} + \mathcal{L}_{grad}}_{\text{Teacher supervision}} + \underbrace{\mathcal{L}_{photo}}_{\text{Novel-view supervision}}. \tag{3}$$

The following sections will explain how we address these issues and design each loss function.

## 3.2 TEACHER-STUDENT MODEL

The aim of the teacher-student model is to solve the scale ambiguity problem during training time to enable single-view 3DGS for the student model with valid 3D geometric structure. Unlike their multi-view counterparts, a single-view model only accepts one view, making it difficult for the model itself to estimate the correct relative scale without cross-view feature matching and triangulation.

**Using the teacher model geometric supervision.** Unlike previous approaches (Nichol et al., 2022; Piccinelli et al., 2024), we do not have access to ground truth 3D annotations. Despite the lack of 3D annotations, during training time, multiple views are provided, and cross-view feature matching can be performed to estimate the Gaussian center for each pixel in the context view with an implicit relative scale (Charatan et al., 2024). Thus, using the teacher model $f_{\boldsymbol{\theta}}^K$, we can convert the dateset from $\{(\boldsymbol{I}^i, \boldsymbol{P}^i)\}_{i=1}^N$ to $\{(\boldsymbol{I}^i, \boldsymbol{P}^i, \boldsymbol{\mu}_t{}^i)\}_{i=1}^N$. Then, in addition to the photometric loss computed from the target view $\{(\boldsymbol{I}^j, \boldsymbol{P}^j)\}_{j=1}^K$, we supervise the student model's Gaussian center predictions $\boldsymbol{\mu}_s^i$ using the teacher's Gaussian centers $\boldsymbol{\mu}_t^i$: $\mathcal{L}_{geo} = \lambda_{geo} \|\boldsymbol{\mu}_t^i - \boldsymbol{\mu}_s^i\|$, where $\|\cdot\|$ is the L1 loss.

**Regularizing local structure consistency.** The L1 loss used in $\mathcal{L}geo$ lacks consideration of the local structure which is prone to distortions in the less confident region such as the boundaries between the in- and out-of context region. To construct good 3D Gaussians and minimize distortions, we also need to encourage consistency in the local structure. Following previous work (Li & Snavely, 2018) that matches the depth map gradients to the ground truth depth map, we match the gradients of the teacher and student Gaussian centers using $\mathcal{L}_{grad} = \lambda_{grad} \|\nabla_{3D} \boldsymbol{\mu}_t^i - \nabla_{3D} \boldsymbol{\mu}_s^i\|$. Unlike previous work (Li & Snavely, 2018) that defines the depth difference between nearby pixels as the gradient map (i.e., only the $z$ value is used for gradient computation), we propose a new definition of gradient $\nabla_{3D}$ that uses the 3D Euclidean distance (i.e., all $x$, $y$, and $z$ values are used for gradient computation) between nearby pixels as the gradient to accommodate 3D structure. This new definition is better aligned with 3D gradient matching.

**Discussion.** The teacher model estimates only the relative scale, and consequently, the student model operates on the same relative scale. The teacher model will not be used at inference time. Therefore, we only require one input view, in other words, we relax the requirement for multiple input views and their corresponding camera poses, to perform the 3D reconstruction, which greatly improves the generalizability. More importantly, the resulting model naturally works as a single-view depth estimation model, connecting the 3D reconstruction task to single-view vision tasks, which goes beyond the capabilities of the teacher model.

## 3.3 EXTRAPOLATION

Unlike multi-view scenarios where the photometric novel view reconstruction loss can be formulated using interpolation only (i.e., enclosing the novel camera view inside the context camera view

frustums), single-view 3D reconstruction inevitably needs to extrapolate when computing the novel view reconstruction loss. This extrapolation can lead to distortion in the extrapolating region as there is no direct visual information. In the case of 3DGS, some 3D Gaussians will be forced to cover the extrapolation region to minimize the photometric loss, which compromises the geometric validity.

**Extrapolating the missing context.** Although the teacher supervision will encourage the Gaussian centers to represent valid geometric shapes, the missing region will create a large photometric loss, which encourages spurious relationships. To minimize this unnecessary photometric loss, we need to either mask out the missing context during loss computation or fill the extrapolating region with additional pixels to avoid noisy gradient flow. We select the latter approach to achieve two functionalities: 1) guide the photometric loss to the correct Gaussians to minimize spurious relationships, and 2) perform extrapolation on the missing context to improve the novel-view reconstruction. We repurpose techniques from (Luo et al., 2018; Rückert et al., 2022) to achieve these functionalities. Instead of directly supervising the rasterized novel view $\hat{\boldsymbol{I}}^j = \mathrm{Rastrizer}(\mathbf{P}^j|\boldsymbol{\mu}^i, \alpha^i, \boldsymbol{\Sigma}^i, \boldsymbol{c}^i)$, we further process the novel-view reconstructions through a network $g_{\boldsymbol{\theta}}^1$ and supervise the output $g_{\boldsymbol{\theta}}^1(\hat{\boldsymbol{I}}^j)$ using the photometric loss $\mathcal{L}_{photo} = \lambda_{l2}\|g_{\boldsymbol{\theta}}^1(\hat{\boldsymbol{I}}^j) - \boldsymbol{I}^j\|_2 + \lambda_{lpips}\mathrm{LPIPS}(g_{\boldsymbol{\theta}}^1(\hat{\boldsymbol{I}}^j), \boldsymbol{I}^j)$, where $\|\cdot\|_2$ is the L2 loss and LPIPS is the Learned Perceptual Image Patch Similarity (Zhang et al., 2018) computed using VGG (Simonyan & Zisserman, 2014) features.

**Using composition to guide gradient flows.** Directly applying $g_{\boldsymbol{\theta}}^1$ will prevent the rasterizer from getting direct supervision, which can harm the reconstruction quality. The ideal situation is to separate the missing context and the visible context using a confidence weight matrix $\boldsymbol{W}$ and treat their losses differently, but this separation is unknown before obtaining the 3D reconstruction. However, we can estimate the missing context using alpha compositing of the 3DGS. More specifically, we construct $\boldsymbol{W}$ by composing the $\alpha^i$. Intuitively, the missing context is less visible and has lower $\alpha^i$ whereas the visible context should have $\alpha^i = 1$. We compose the novel view as $\hat{\boldsymbol{I}}_c^j = g_{\boldsymbol{\theta}}^1(\hat{\boldsymbol{I}}^j) \odot (\mathbf{1} - \boldsymbol{W}^j) + \hat{\boldsymbol{I}}^j \odot \boldsymbol{W}^j$, where $\boldsymbol{W}^j = \mathrm{Rastrizer}(\mathbf{P}^j|\boldsymbol{\mu}^i, \alpha^i, \boldsymbol{\Sigma}^i, \mathbf{1})$. Then, we can guide the gradients computed from $\mathcal{L}_{photo} = \lambda_{l2}\|\hat{\boldsymbol{I}}_c^j - \boldsymbol{I}^j\|_2 + \lambda_{lpips}\mathrm{LPIPS}(\hat{\boldsymbol{I}}_c^j, \boldsymbol{I}^j)$ for the missing context to the extrapolation network, but the gradients for the context to the rasterizer, and the rasterizer always gets direct supervision from the reconstruction loss. Additionally, the existence of $\boldsymbol{W}$ allows the student model to balance between the completeness and the confidence of the reconstruction by generating lower opacity for the regions with less confidence, since $g^1\boldsymbol{\theta}$ can still fill the less opaque area to minimize the loss. On the other hand, $\boldsymbol{W}$ cannot collapse to zero, as $g^1\boldsymbol{\theta}$ will not be able to fill anything without context. Finally, the learned $\boldsymbol{W}$ can be used during inference to identify the missing context.

**Discussion.** Better extrapolation networks, such as diffusion-based methods (lkwq007, 2023), can be applied to achieve better novel-view reconstruction quality, but they make the training pipeline more complicated. We choose a feed-forward network (i.e., a pre-trained GAN network) to match the base training pipeline and preserve efficiency. The main goal of the extrapolator here is to direct the gradient flow to minimize artifacts. The ability to learn $\boldsymbol{W}$ is more important than generating the best extrapolation result; as long as some level of extrapolation can be achieved and the learned context mask $\boldsymbol{W}$ is accurate, we can apply more elaborate extrapolation methods, such as differential diffusion (Levin & Fried, 2023), during inference using the generated context mask. We visualize $\boldsymbol{W}$ in the Appendix C. Because of the introduction of the extrapolator, we can use the student network to produce additional views by providing fake camera poses. This way, assuming the teacher model performs better than the student model, we can use the teacher model to process the student model's output views to further improve the reconstruction result. This setting is detailed in the Appendix D.

## 4 EXPERIMENTS

### 4.1 SETTINGS

**Datasets.** To evaluate the novel-view reconstruction performance, we follow previous multi-view approaches (Charatan et al., 2024; Chen et al., 2024) by using RealEstate10k (RE10k) (Zhou et al., 2018) and ACID (Liu et al., 2021). These two datasets contain multiple views and the corresponding camera poses generated using a Structure from motion algorithm (Schönberger & Frahm, 2016) for different indoor and outdoor scenes. To evaluate the geometric quality and the potential to serve

| Method | Views (#) | Params (M) | RE10K (Zhou et al., 2018) | | | ACID (Liu et al., 2021) | | |
|---|---|---|---|---|---|---|---|---|
| | | | PSNR↑ | SSIM↑ | LPIPS↓ | PSNR↑ | SSIM↑ | LPIPS↓ |
| *Interpolation* pixelNeRF (Yu et al., 2021) | 2 | 28.2 | 20.43 | 0.589 | 0.550 | 20.97 | 0.547 | 0.533 |
| GPNR (Suhail et al., 2022) | 2 | 9.6 | 24.11 | 0.793 | 0.255 | 25.28 | 0.764 | 0.332 |
| AttnRend (Du et al., 2023) | 2 | 125.1 | 24.78 | 0.820 | 0.213 | 26.88 | 0.799 | 0.218 |
| MuRF (Xu et al., 2024) | 2 | 5.3 | 26.10 | 0.858 | 0.143 | 28.09 | 0.841 | 0.155 |
| pixelSplat (Charatan et al., 2024) | 2 | 125.4 | 25.89 | 0.858 | 0.142 | 28.14 | 0.839 | 0.150 |
| MVSplat (Chen et al., 2024) | 2 | 12.0 | 26.39 | 0.869 | 0.128 | 28.25 | 0.843 | 0.144 |
| *Extrapolation* pixelSplat (Charatan et al., 2024) | 2 | 125.4 | 24.20 | 0.843 | 0.162 | 27.38 | 0.838 | 0.157 |
| MVSplat (Chen et al., 2024) | 2 | 12.0 | 23.48 | 0.834 | 0.163 | 26.39 | 0.831 | 0.158 |
| pixelSplat (Charatan et al., 2024) | 1 | 125.4 | 20.15 | 0.662 | 0.256 | 23.40 | 0.670 | 0.242 |
| MVSplat (Chen et al., 2024) | 1 | **12.0** | 17.73 | 0.585 | 0.296 | 20.17 | 0.581 | 0.288 |
| SplatterImage (Szymanowicz et al., 2024) | 1 | 62.1 | 22.32 | 0.754 | 0.197 | 25.08 | 0.738 | 0.204 |
| studentSplat | 1 | 32.0 | **24.98** | **0.794** | **0.156** | **26.94** | **0.767** | **0.160** |

Table 1: **Novel-view reconstruction performance**. The best performance in the single-view setting is bold, the second is underlined. The original interpolation performance are included for reference.

as a self-supervised depth estimation method, we use the indoor and outdoor annotations from DA-2K (Yang et al., 2024) and DIODE (Vasiljevic et al., 2019).

**Metrics.** The novel-view reconstruction performance is evaluated using photometric metrics, including pixel-level Peak Signal-To-Noise Ratio (PSNR), patch-level Structural Similarity Index Measure (SSIM) (Wang et al., 2004), and feature-level Learned Perceptual Image Patch Similarity (LPIPS) (Zhang et al., 2018). The depth estimation metrics follow standard practice by using Absolute Relative Error (AbsRel), $\delta_1$, and accuracy on the corresponding datasets. All experiments are performed using $256 \times 256 \times K$, where $K$ is the number of views. Thus, single-view methods have lower resolution. The evaluation settings are detailed in the Appendix B.

**Implementation Details.** Since our goal is to design a new proof-of-concept approach instead of improving current ones, we aim for a balance between performance and efficiency rather than absolute performance. We expect larger models to produce better results. We use an efficient method, MVSplat (Chen et al., 2024), as the teacher model. For the student model, we combine the DINOv2 (Oquab et al., 2023) pre-trained ViT-S backbone with the DPT (Ranftl et al., 2021) head as the architecture, as it has been shown to perform well in single-view depth estimation. Following MVSplat, we use a shallow ResNet (He et al., 2016) encoded features and the original images to re-fine the output depth map. For the extrapolator, we use the efficient MI-GAN (Sargsyan et al., 2023) inpainter. Additional details and results from different encoders are provided in the Appendix C.

## 4.2 QUANTITATIVE COMPARISONS

**Novel-view reconstruction performance.** To perform a quantitative comparison with the current state-of-the-art (SOTA) methods on 3D scene reconstruction performance without 3D annotations, we follow previous work to evaluate the novel-view reconstruction. Additionally, we aim to evaluate the extrapolation capability. Therefore, unlike previous work that only bounded the novel views by the context view frustums, we also evaluate the reconstruction performance using views both inside and outside the context view frustums. As suggested by previous work (Charatan et al., 2024) that current scene-level 3DGS methods cannot perform extrapolation, we can see from Table 1 that the performance of a SOTA multi-view 3DGS method drops when performing extrapolation. Additionally, a single-view 3DGS method, SplatterImage (Szymanowicz et al., 2024), outperforms the SOTA multi-view 3DGS method when only one view is provided, which suggests that multi-view 3DGS methods cannot be directly applied to the single-view setting despite their promising performance in the multi-view setting; directly training the single-view method will result in better reconstruction performance. This result supports the necessity of training a single-view 3DGS method. Furthermore, our studentSplat achieves the best single-view 3DGS performance and is on par with the multi-view models despite using only one input view, which demonstrates the effectiveness of the teacher-student architecture and extrapolation capability. However, we acknowledge that our SSIM score is still behind the multi-view methods. The inferior performance can be partially attributed to the resolution difference. All the methods generate one 3D Gaussian for each image pixel; the meth-

ods using two input views have twice the number of 3D Gaussians to render from, thus resulting in a sharper image, which in turn results in a higher SSIM score.

| Method | Views (#) | ACID (Liu et al., 2021) | | | DTU (Aanæs et al., 2016) | | |
|---|---|---|---|---|---|---|---|
| | | PSNR↑ | SSIM↑ | LPIPS↓ | PSNR↑ | SSIM↑ | LPIPS↓ |
| pixelSplat (Charatan et al., 2024) | 2 | 27.64 | 0.830 | 0.160 | 12.89 | 0.382 | 0.560 |
| MVSplat (Chen et al., 2024) | 2 | 28.15 | 0.841 | 0.147 | 13.94 | 0.473 | 0.385 |
| MVSplat (Chen et al., 2024) | 1 | 21.13 | 0.631 | 0.261 | 9.67 | 0.245 | 0.602 |
| SplatterImage (Szymanowicz et al., 2024) | 1 | 24.95 | 0.735 | 0.200 | 12.39 | 0.353 | 0.542 |
| studentSplat | 1 | **26.59** | **0.758** | **0.167** | **14.15** | **0.411** | **0.491** |

Table 2: **Cross-dataset generalization in novel view reconstruction.** Results from models trained on RealEstate10K. The best performance in singel-view novel-view reconstruction is bold and the second is underlined. The original multi-view interpolation results are included for reference.

**Novel-view reconstruction generalizability.** We follow MVSplat (Chen et al., 2024) to evaluate the cross-dataset novel-view reconstruction performance. As shown in Table 2, MVSplat again does not work in the single-view setting. On the other hand, our studentSplat achieved the best single-view performance and is on par with multi-view pixelSplat, depending on the dataset. This result further supports the effectiveness of our method and shows the potential for our method to act as a generalizable single-view vision encoder.

| Method | DIODE (Vasiljevic et al., 2019) | | DA-2K (Yang et al., 2024) |
|---|---|---|---|
| | $\delta_1$↑ | AbsRel↓ | Acc (%)↑ |
| GasMono (Zhao et al., 2023) | 0.504 | **0.348** | 0.700 |
| SplatterImage (Szymanowicz et al., 2024) | 0.395 | 1.457 | 0.615 |
| studentSplat | **0.604** | 0.407 | **0.708** |

Table 3: **Cross-dataset generalization in self-supervised single-view depth estimation.** Splatter-Image and studentSplat are trained on RealEstiate10K. GasMono is taken from the original work. Testing dataset unseen during training.

**Self-supervised single-view depth estimation performance.** We evaluate the single-view depth estimation performance of our method against a SOTA self-supervised single-view depth estimation method, GasMono (Zhao et al., 2023), and a SOTA single-view object 3DGS model, SplatterImage (Szymanowicz et al., 2024). Note that the evaluation datasets are unseen by any of the models. From Table 3, we see that our method achieved much better performance than SplatterImage and on-par performance with GasMono. This result further supports the generalizability of our method and the potential to serve as a self-supervised single-view depth estimation method.

## 4.3 QULITATIVE COMPARISONS

In this section, we aim to visualize the proposed studentSplat in terms of extrapolation performance, distortion, and reconstruction quality. The qualitative comparisons for depth estimation and integration with Stable Diffusion (Rombach et al., 2022) for text-to-3D generation are in the Appendix C.

**Better extrapolation performance.** Thanks to our extrapolator, our studentSplat is able to fill the missing context, as shown in Figure 3, whereas previous methods leave the region blank or stretch the border Gaussians to fill the region.

**Less distortion compared to current single-view methods.** From the last two columns of Figure 3, we see that SplatterImage tends to create a jelly effect around the border of the context, which is the distortion we aim to minimize, and our method does not have such distortion.

**Similar reconstruction quality with less resolution.** Since our studentSplat uses one input view instead of two views, we generate half the number of 3D Gaussians (i.e., half the resolution). Despite the lower resolution and sharpness, our studentSplat still generates overall comparable reconstructions to the multi-view (higher resolution) methods, as shown in Figure 3.

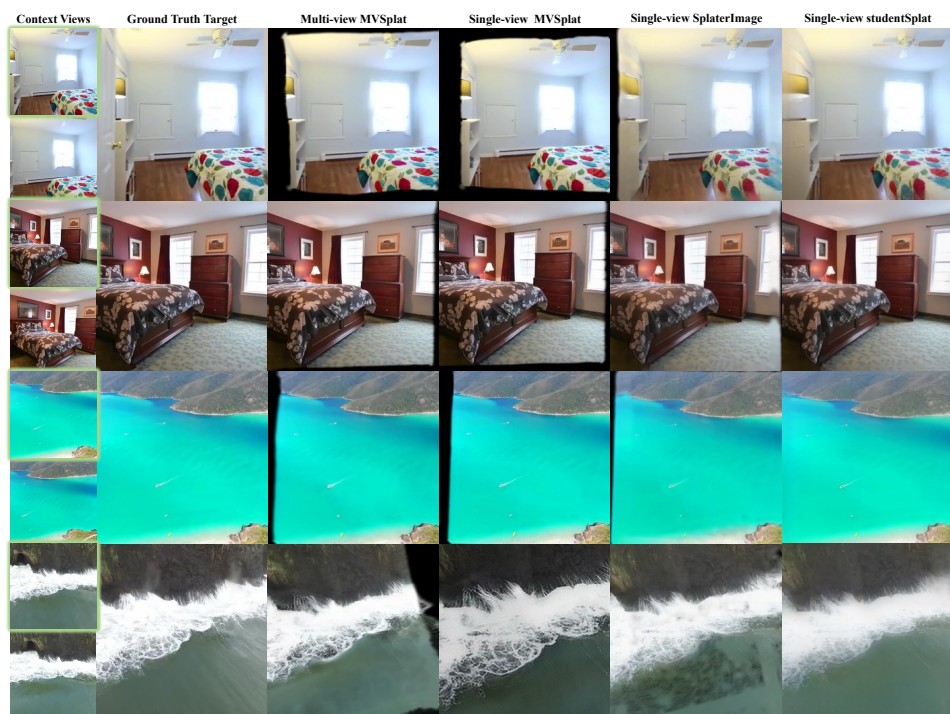

Figure 3: The qualitative comparison between representative methods in the extrapolation setting. Top two rows are from RE10K and the bottom two rows are from ACID The multi-view method use both of the context views whereas the single-view method only use the context view highlighted in green. Additional examples are in the Appendix C.

**Generalizable reconstruction quality.** The advantage of studentSplat generalizes to unseen domains. As shown in Figure 4, our method is able to complete the missing region with low distortion. However, due to the lower resolution (i.e., fewer 3D Gaussians), our results are less sharp.

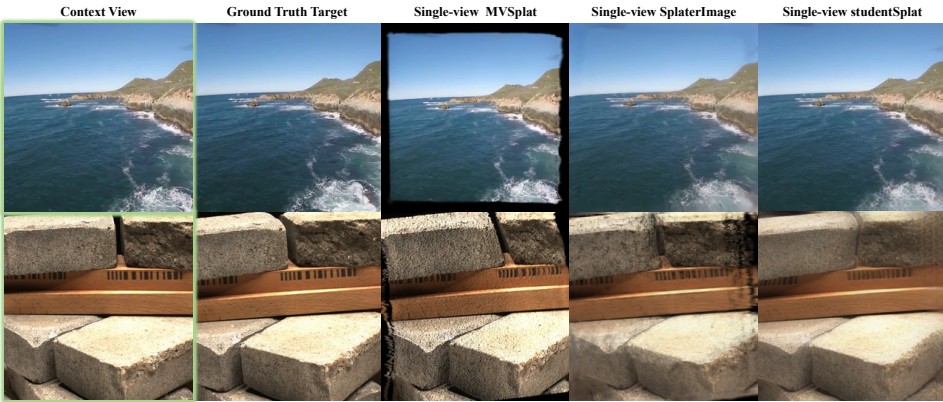

Figure 4: The qualitative comparison between representative methods in the single-view cross-dataset generalization setting. The context view is highlighted in green.

## 4.4 ABLATION STUDY

In this ablation study, we aim to evaluate how each of the proposed modules contributes to the model's performance by iteratively removing the proposed modules.

**Ablation on the extrapolator.** From Table 4, we observe that removing composition results in a slight performance drop across metrics. If we additionally remove the extrapolator, we have a large performance drop, which demonstrates the necessity of the extrapolator.

**Ablation on the teacher geometric supervision.** The quantitative measurements alone will be misleading for ablating the geometric supervision, as photometric measurements do not consider geometric validity. Therefore, we evaluate the modules both quantitatively and qualitatively. We see from Table 4 that removing gradient matching improves PSNR but reduces LPIPS. This only makes sense when considering Figure 5, where removing gradient matching results in a large number of Gaussians being misplaced in the missing region. Although this misplacement improves the PSNR score, it lowers the structure validity which we aim to preserve. If we additionally remove the entire teacher geometric supervision, we observe that the reconstruction performance improves, which seems counterintuitive. However, we can see from Figure 5 that the improvements again come with sacrificing the geometric validity; the jelly distortion similar to SplatterImage appears, which even affects the in-context region. Additional ablations in depth estimation performance are in the Appendix C. These results demonstrate the effectiveness of the proposed modules.

| Ablation Module | Setup | PSNR↑ | SSIM↑ | LPIPS↓ |
|---|---|---|---|---|
| | Final | 24.98 | 0.794 | 0.156 |
| Extrapolator | +w/o Composition | 24.85 | 0.792 | 0.158 |
| | +w/o Extrapolation | 21.38 | 0.741 | 0.208 |
| Supervision | +w/o Gradient Matching | 21.57 | 0.741 | 0.211 |
| | +w/o Teacher | 22.13 | 0.757 | 0.195 |

Table 4: **Ablations on RealEstate10K**. We separate the ablation into Extrapolator where we ablate the components in the extrapolator and Supervision where we ablate the geometric supervision loss.

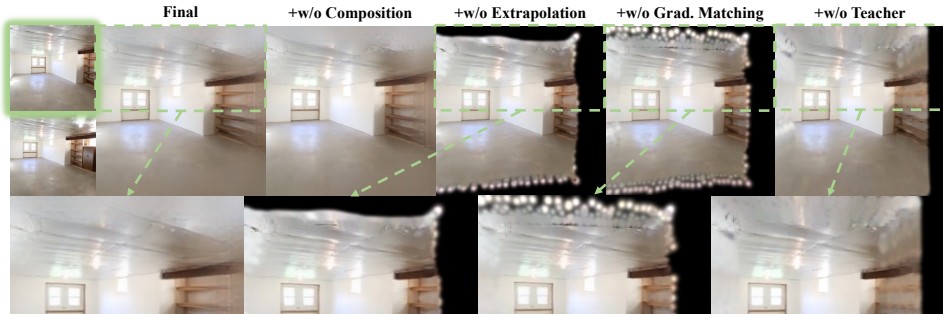

Figure 5: **The qualitative ablation results.** The input view is highlighted in green. The ground truth target view is below the input view. We zoomed in some areas for better comparison.

## 4.5 DISCUSSION AND CONCLUSION

We demonstrate, using studentSplat, the possibility of single-view 3DGS at scene level, bridging the gap between 3DGS and single-view depth estimation. With its modular design, studentSplat allows for versatile applications (see Appendix C and D) and easy incorporation of better modules.

**Limitations and future direction.** Our method relies on the teacher model, thus inheriting the limitations of the teacher model. It would be interesting to eliminate the need for the teacher model to further improve the capability of single-view scene-level 3DGS. Additionally, training a single-view 3DGS model is still more difficult than training its multi-view counterparts, so our method cannot outperform the multi-view method in its current stage. Large-scale training is an interesting direction to explore the capability of single-view 3DGS for both novel-view reconstruction and depth estimation tasks. Furthermore, we expect our method to also aid other vision tasks like semantic segmentation, which can be another direction to explore. Finally, as a proof-of-concept approach, many design aspects, such as model architecture and loss function design, can be optimized.

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

## A    ADDITIONAL IMPLEMENTATION DETAILS

**Student Architecture.** The student network architecture is shown in Figure 6. It only requires the images as input (i.e., without camera pose requirements). It comprises a backbone branch and a refine branch, similar to previous work (Charatan et al., 2024; Chen et al., 2024). The backbone branch localizes the Gaussian centers along the $z$-axis, whereas the refine branch uses CNN features and input images to refine the backbone prediction and predict other Gaussian parameters.

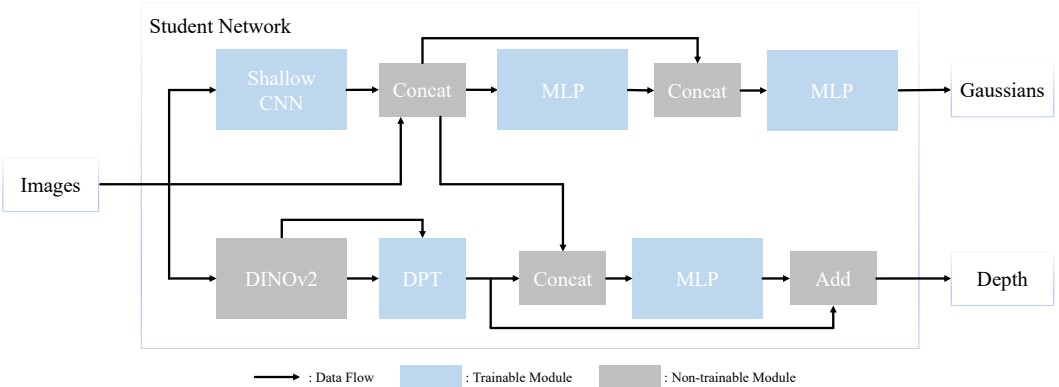

Figure 6: **Student network architecture.** The shallow CNN is the same as previous work (Chen et al., 2024) but randomly initialized. The MLP conposes of a 3x3 Conv, a GeLU (Hendrycks & Gimpel, 2016) activation, and a 1x1 Conv.

**Novel-view reconstruction.** To generate a novel-view, our studentSplat first generates the novel views directly using the rendering function from 3DGS. Additionally, we use the rendering function to generate the opacity map. The novel-view renderings and the opacity map are processed by the extrapolator to generate the complete novel views.

**Depth normalization.** To learn a generalizable depth map, we use the provided camera intrinsics, near plane, and far plane to scale, shift, and clip the predicted depth map, respectively, using: $\mathrm{depth\_scaled} = \mathrm{Max}(\mathrm{focal\_length} * \mathrm{depth} + \mathrm{near}, \mathrm{far})$

**More training details.** All our models are trained on two A10G GPUs with a total batch size of 2 for 300,000 iterations with the Adam (Kingma & Ba, 2014) optimizer. Each batch contains one training scene (i.e., two input views and four target views). For all experiments, we use an initial learning rate of 2e-4 and a cosine learning rate scheduler with 2000 warm-up iterations. All the models are trained for 300,000 iterations. Same as MVSplat (Chen et al., 2024), the frame distance between two input views is gradually increased as the training progresses. For both RE10K (Zhou et al., 2018) and ACID (Liu et al., 2021), we follow previous works (Charatan et al., 2024; Chen et al., 2024) to set the near and far depth planes to 1 and 100, respectively. For DTU (Aanæs et al., 2016), we use the provided near and far depth planes of 2.125 and 4.525, respectively.

## B    EVALUATION SETTINGS

**Novel-view reconstruction.** For the interpolation setting, we use the reported numbers from previous work for reference. Those evaluations are done using 3 novel views inside the context frustums. In our extrapolation setting, we use 2 novel views outside and one novel view inside the context frustums. All the multi-view methods use all the context views to produce the 3D Gaussians. Comparing single-view methods to multi-view methods is inherently unfair since single-view methods have less information and lower resolutions (i.e., fewer 3D Gaussians). Although we cannot avoid this unfairness, to better compare multi-view and single-view methods, we use the context view that produces the best SSIM score for each target view as the input for the single-view method. It is not intuitive to apply the multi-view methods (i.e., pixelSplat (Charatan et al., 2024) and MVSplat (Chen et al., 2024)) in the single-view setting. To adapt them, we simply repeat the input view to create another view, since the training data already contains views that are very close to each other. We noticed in

the GitHub issue https://github.com/donydchen/mvsplat/issues/37 that we may warp the input view to create a fake view. However, this is impossible without the scale or depth information.

**Depth estimation.** DA2K (Yang et al., 2024) is annotated by human on depth relationship between two pixels (i.e., which pixel is closer). To make sure both pixels are on the same image and to keep the aspect ratio, we pad the shorter edge of the image to the longer edge size and resize to $256 \times 256$. DIODE(Vasiljevic et al., 2019) dataset has the ground truth depth map with mask, we first extract two square crops from each image with maximum coverage and resize each crop to $256 \times 256$. Next, we perform median scaling to both the predicted depth map and the ground truth depth map. Then, we apply the mask on both the predicted depth map and the ground truth depth map before computing the metrics. Finally, we average the metrics over all the crops.

**Depth estimation metric.** The metrics are defined following previous work (Eigen et al., 2014). More specifically, the AbsRel, the absolute value of the difference between predicted depth and ground truth depth relative to the ground truth depth, and $\delta_1$, the percentage of pixel with predicted depth close enough to the ground truth depth, are defined as:

$$\mathrm{AbsRel}(\hat{\boldsymbol{D}}, \boldsymbol{D}) = \frac{1}{\|\boldsymbol{D}\|} \sum_{\hat{d}, d \in \hat{\boldsymbol{D}}, \boldsymbol{D}} |\hat{d} - d|/d, \tag{4}$$

$$\delta_1(\hat{\boldsymbol{D}}, \boldsymbol{D}) = \frac{1}{\|\boldsymbol{D}\|} \|\{\hat{d}, d \in \hat{\boldsymbol{D}}, \boldsymbol{D} | \mathrm{Max}(\frac{\hat{d}}{d}, \frac{d}{\hat{d}}) < 1.25\}\|, \tag{5}$$

where $|\cdot|$ is the absolute value, $\|\cdot\|$ is the size of a matrix of the cardinality of a set, $\boldsymbol{D}$ is the ground truth depth map, $\hat{\boldsymbol{D}}$ is the predicted depth map, and $\hat{d}, d \in \hat{\boldsymbol{D}}, \boldsymbol{D}$ represents taking the depth values $\hat{d}, d$ from each matrix at the corresponding pixels.

## C  MORE RESULTS

**Encoder without large-scale pre-training.** We also trained our model on RE10K dataset using DINO (Caron et al., 2021) with ImageNet (Russakovsky et al., 2015) pre-trained weights (i.e., one tenth of the training data compare to DINOv2 (Oquab et al., 2023)) to evaluate how much the pre-trained encoder contributes to our model performance. From Table 5, we observe a performance drop without using large-scale pre-trained weights which is expected. However, the performance drop is much smaller compare to model trained without the proposed modules. Therefore, the proposed modules are the main contributor to studentSplat's performance.

| Setup | PSNR↑ | SSIM↑ | LPIPS↓ |
|---|---|---|---|
| Final | 24.98 | 0.794 | 0.156 |
| +w/o Extrapolation | 21.38 | 0.741 | 0.208 |
| +w/o Teacher | 22.13 | 0.757 | 0.195 |
| w/o Large-scale Pre-train | 24.63 | 0.783 | 0.163 |

Table 5: Compare novel view reconstruction results w/ and w/o large-scale pre-trained encoder on RealEstate10K

**Student model with ground truth depth pre-training.** We also trained our model on the RE10K dataset using pre-trained weights from Depth Anything V2 (Yang et al., 2024) (i.e., the training data contains ground truth depth labels) to evaluate if we can enhance the reconstruction quality when the student model has prior depth knowledge. From Table 6, we observe a performance improvement using Depth Anything V2 weights, which suggests that the performance of our studentSplat can be further improved if we employ a depth estimation model as the student model. This result further reinforces the connection between depth estimation and 3DGS.

**Depth estimation and teacher supervision.** In addition to the ablation results in the main text, we validate the effectiveness of teacher supervision on geometric validity by performing depth estimation. As shown in Table 7, the method without gradient matching performs worse, and the model

| Setup | PSNR↑ | SSIM↑ | LPIPS↓ |
|---|---|---|---|
| Final | 24.98 | 0.794 | 0.156 |
| +w/ Depth Anything V2 Weights | 25.11 | 0.798 | 0.154 |

Table 6: Compare novel view reconstruction results w/ and w/o Depth Anything V2 (Yang et al., 2024) weights

without teacher supervision suffers a significant performance drop. These results further validate the effectiveness of the proposed teacher supervision.

| Method | DIODE (Vasiljevic et al., 2019) | | DA-2K (Yang et al., 2024) |
|---|---|---|---|
| | $\delta_1$↑ | AbsRel↓ | Acc (%)↑ |
| GasMono (Zhao et al., 2023) | 0.504 | 0.348 | 0.700 |
| SplatterImage (Szymanowicz et al., 2024) | 0.395 | 1.457 | 0.615 |
| Final | 0.604 | 0.407 | 0.708 |
| +w/o Gradient Matching | 0.606 | 0.413 | 0.683 |
| +w/o Teacher | 0.541 | 1.526 | 0.653 |

Table 7: Cross-dataset generalization in self-supervised single-view depth estimation w/ and w/o teacher supervision.

**Qualitative results.** Additional novel-view reconstructions are shown in Figure 8. The extrapolating region have lower quality and different content compare to the ground truth. Single-view results can be slightly less sharp. Qualitative results of self-supervised single-view depth estimation are visualized in Figures 9, 10, and 11 for the DA2K (Yang et al., 2024), DIODE indoor (Vasiljevic et al., 2019), and DIODE outdoor (Vasiljevic et al., 2019) datasets, respectively. Our studentSplat produces less noise compared to SplatterImage (Szymanowicz et al., 2024) and is comparable to GasMono (Zhao et al., 2023). The context confidence weight matrix $W$ is visualized in Figure 7. The darker regions are less confident, while the brighter regions are more confident. We also show the thresholded $W$ at 0.5 for better visualization. Note that the model is less confident at regions with missing information and object boundaries, where missing context from occlusion tends to happen. This confidence weight guides the extrapolator in our studentSplat model.

**Scene-level text-to-3D generation pipeline.** Generating new 3D views is helpful for 3D design and content creation. Current methods on text-to-3D scene generation require per-scene optimization (Zhang et al., 2024a), multiple iterations and depth refinement (Ouyang et al., 2023; Fridman et al., 2024; Zhang et al., 2024b), or are constrained by predefined objects (Chang et al., 2014). By combining studentSplat with Stable Diffusion (Rombach et al., 2022), we can produce a scene-level text-to-3DGS method that generates diverse 3D scenes without depth guidance. More importantly, we can obtain a text-to-3D scene pipeline without training a 3D generative model. The results are shown in Figure 12. We apply a fake forward camera shift of 0.2 and use the intrinsics from the training data.

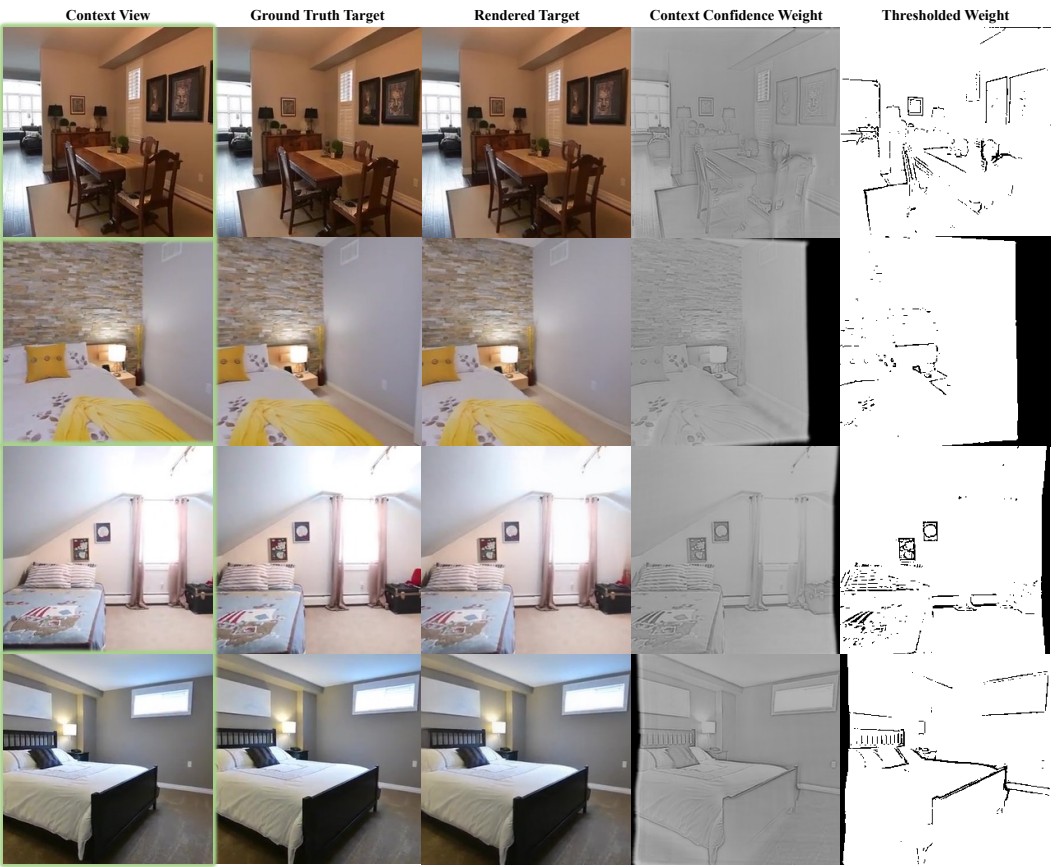

| Context View | Ground Truth Target | Rendered Target | Context Confidence Weight | Thresholded Weight |

Figure 7: Visualization of the context confidence weight $W$ on RE10K dataset. Our studentSplat is more confident at the brighter regions and less confident at the darker regions. The less confident regions of the rendered target are complete by the extrapolator.

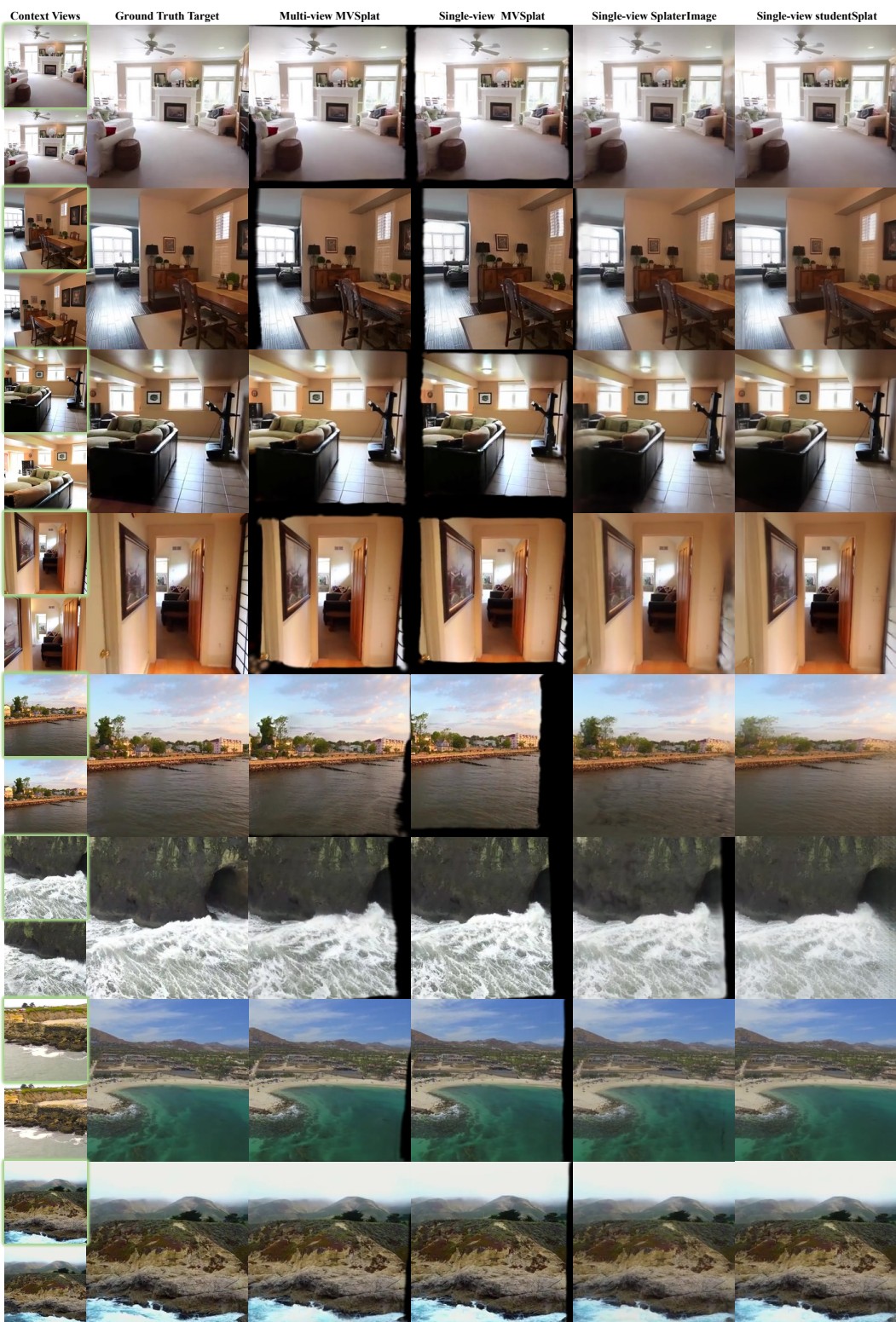

Figure 8: Additional qualitative comparison between representative methods in the extrapolation setting. The top four rows are from RE10K, and the bottom four rows are from ACID. The multi-view method uses both context views, whereas the single-view method only uses the context view highlighted in green.

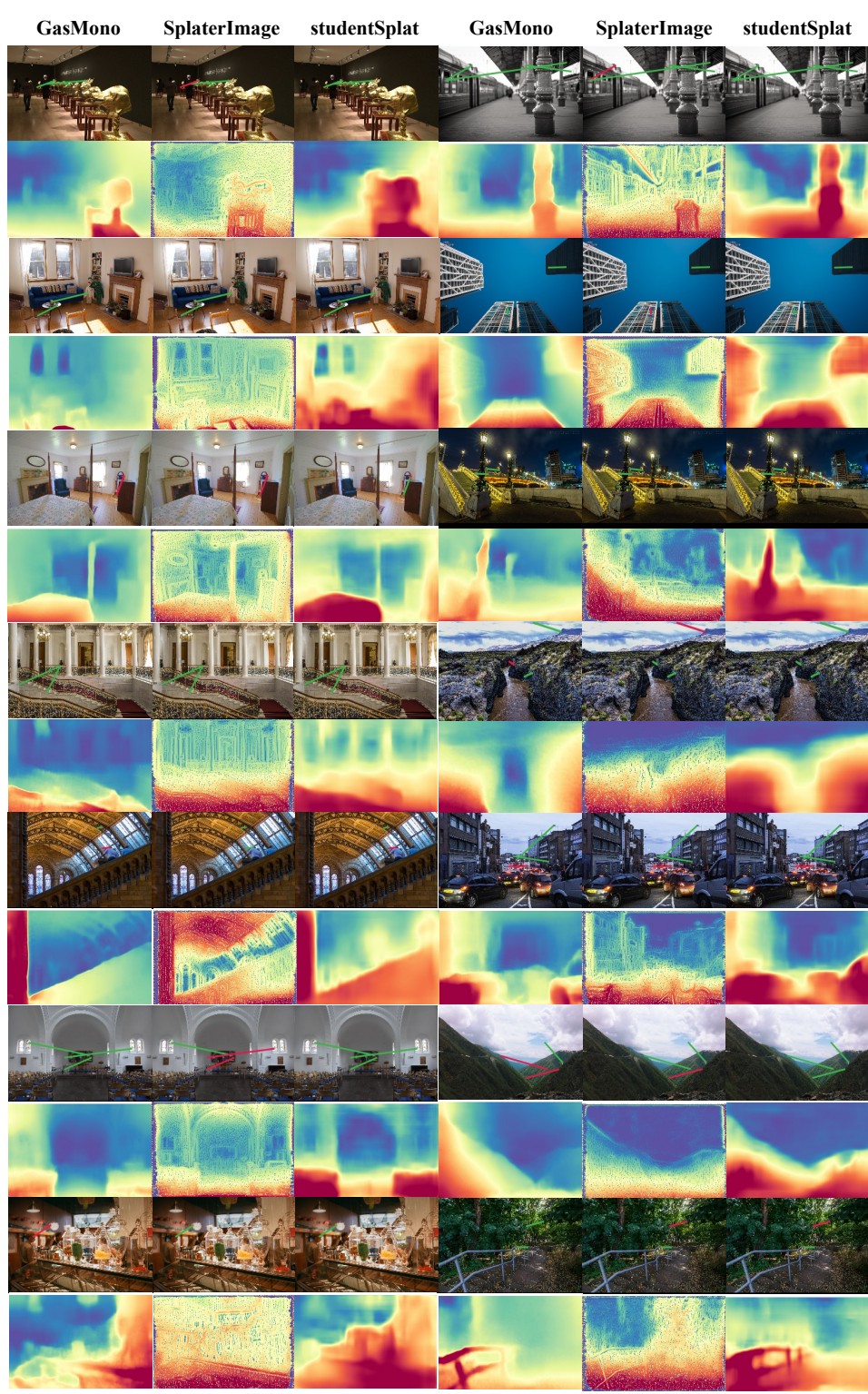

Figure 9: Additional qualitative comparison between representative methods for self-supervised single-view depth estimation performance on the DA2K dataset. Line segments in the original images represent the predicted depth difference (red: incorrect, green: correct).

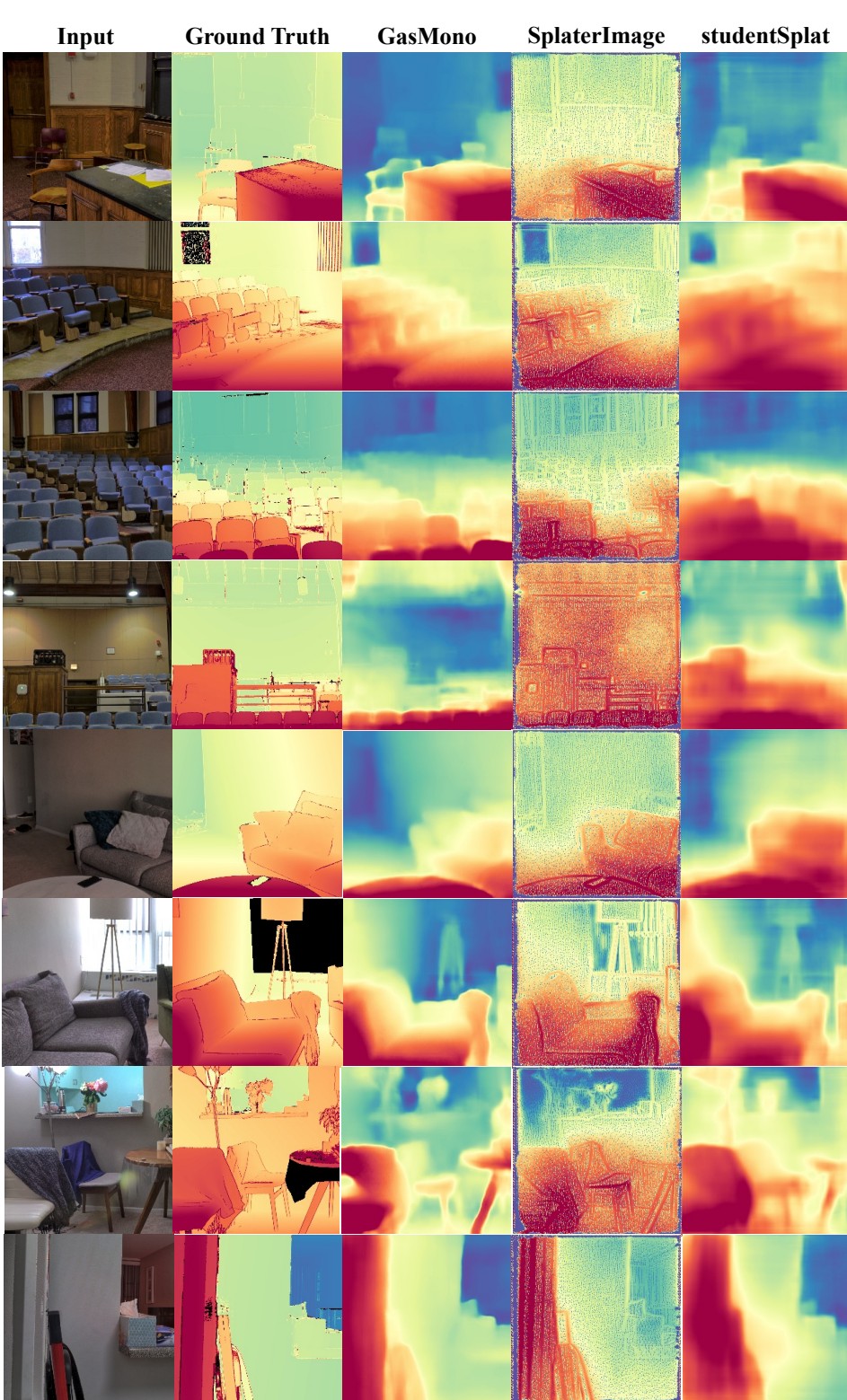

Figure 10: Additional qualitative comparison between representative methods for self-supervised single-view depth estimation performance on the DIODE indoor dataset.

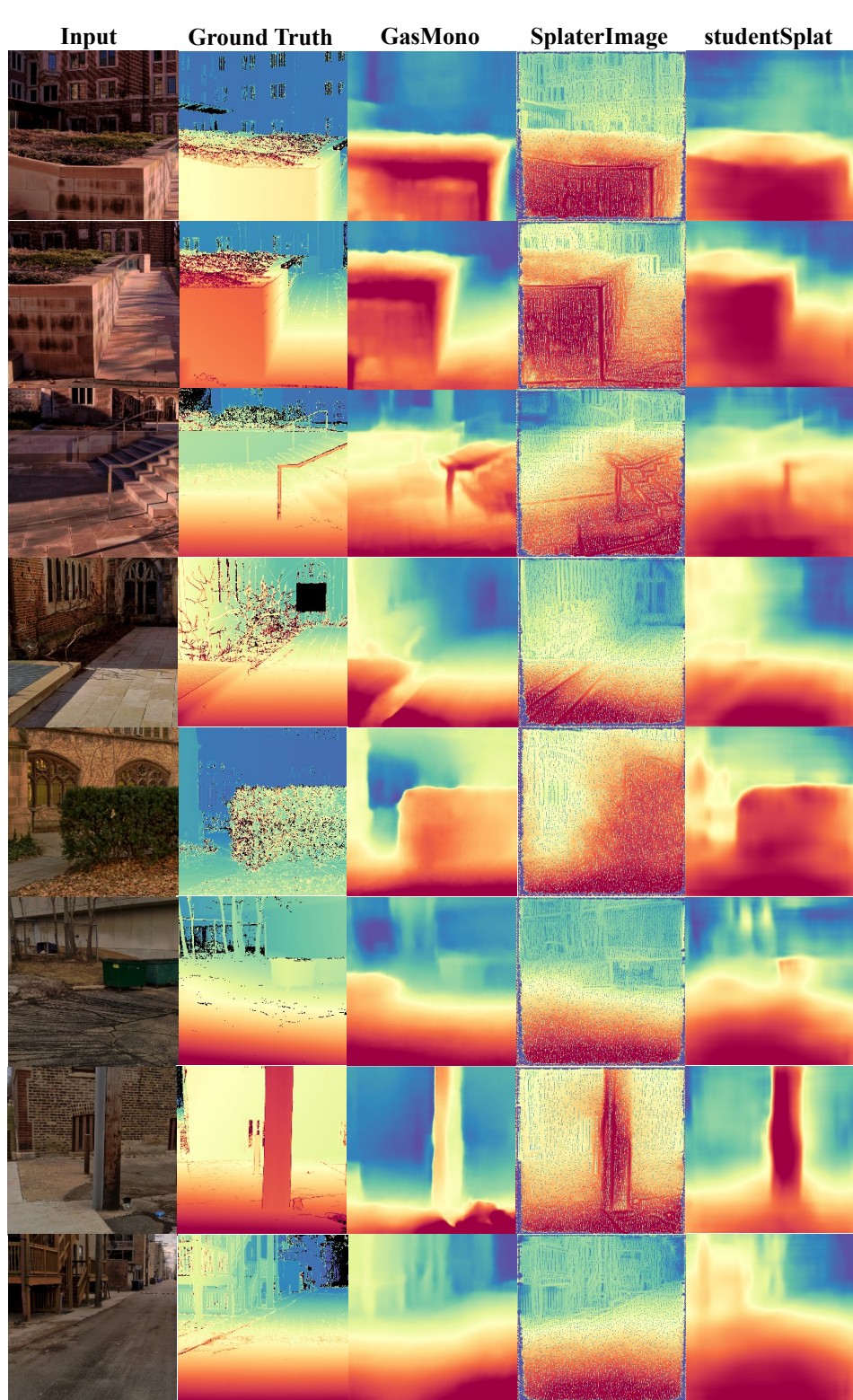

Figure 11: Additional qualitative comparison between representative methods for self-supervised single-view depth estimation performance on the DIODE outdoor dataset.

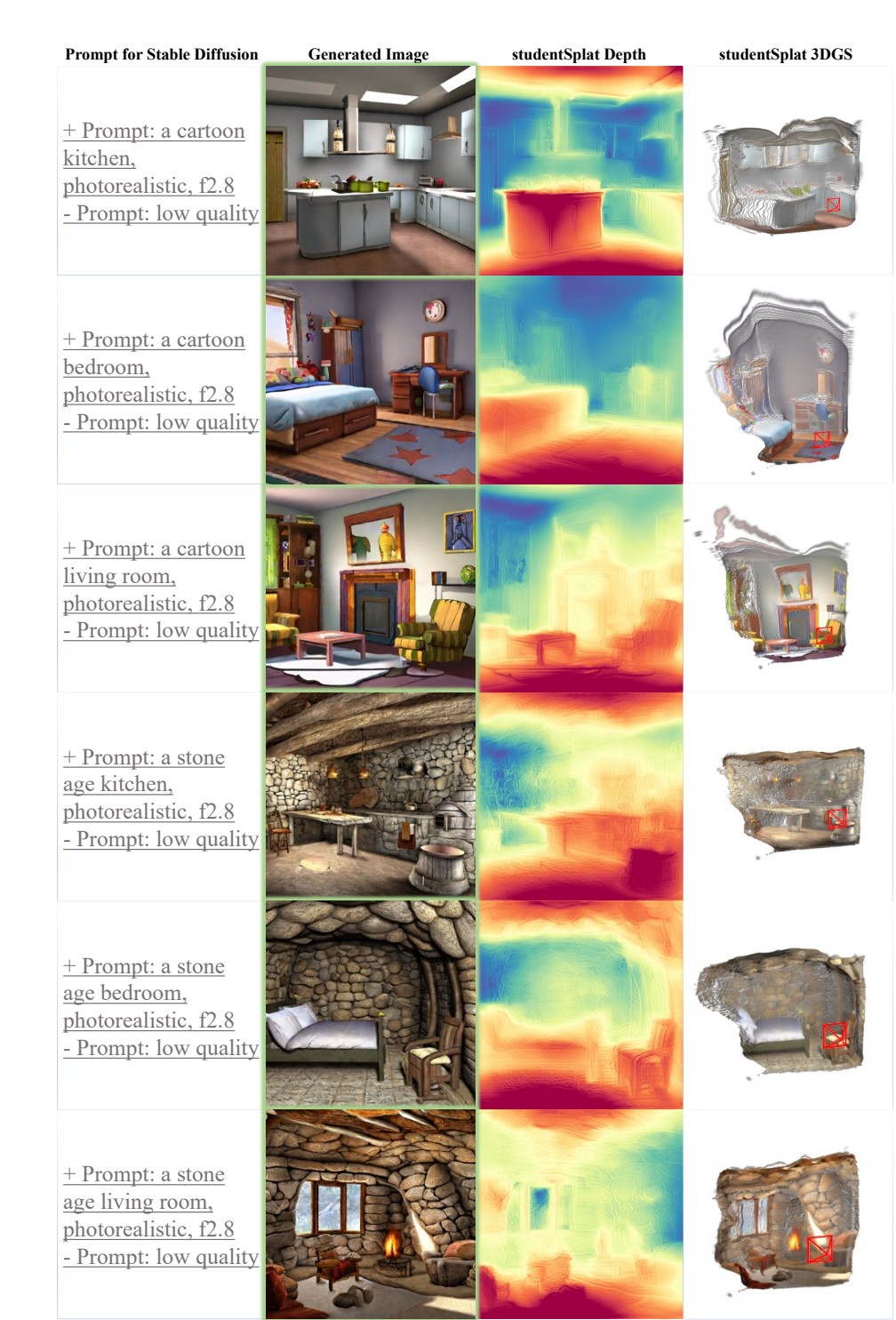

Figure 12: Visualization of the text-to-3D generation result using studentSplat with teacher refine detailed in Section D on the Stable Diffusion output. The input is highlighted in green.

# D  REFINING STUDENT OUTPUT WITH TEACHER MODEL

## D.1  METHOD

The single-view studentSplat generally produces better quality novel-view reconstructions and extrapolation when the camera view change is small. More importantly, the multi-view teacher model still performs better in 3D reconstruction than the student model. These properties lead us to another design that further improves the 3D reconstruction performance. Specifically, we use studentSplat to generate good quality novel views using one input view and fake camera poses with small shifts. Then, the input view and generated novel views with the fake camera poses are used as the context for the teacher input. The advantage of this pipeline is that we preserve the single-view nature of our studentSplat and only trade off the inference speed for performance improvements. The overall pipeline is shown in Figure 13.

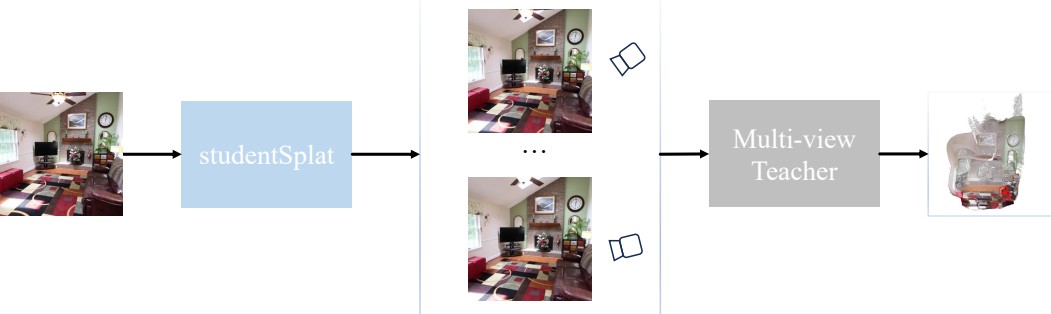

Figure 13: **The pipeline to refine the student output with the teacher model.** The student model generates additional viewpoints using user-specified virtual camera poses. The teacher model utilizes these generated viewpoints and the corresponding virtual camera poses to refine the camera pose estimates.

## D.2  RESULTS

Using the teacher refinement, we can improve the quality of the generated 3D structure. We show the improvements using the single-view depth estimation task. We use a forward ($z$-axis) shift of 0.5 to produce the relative camera poses. All the camera intrinsics, near plane, and far plane are directly taken from the training dataset RE10K (Zhou et al., 2018). Only one image is provided to the pipeline to predict the depth. As shown in Table 8, the additional use of teacher refinement results in noticeable performance improvements. We can also see from Figure 14 that the refined depth maps are much sharper.

| Method | DIODE (Vasiljevic et al., 2019) | | DA-2K (Yang et al., 2024) |
| --- | --- | --- | --- |
| | $\delta_1\uparrow$ | AbsRel$\downarrow$ | Acc (%)$\uparrow$ |
| GasMono (Zhao et al., 2023) | 0.504 | **0.348** | 0.700 |
| studentSplat | 0.604 | 0.407 | 0.708 |
| studentSplat w/ teacher refine | **0.623** | 0.397 | **0.716** |

Table 8: **Cross-dataset generalization in self-supervised single-view depth estimation.** The studentSplat is trained on the RealEstate10K dataset. "Teacher refine" refers to the additional use of the teacher network to refine the output of the student model.

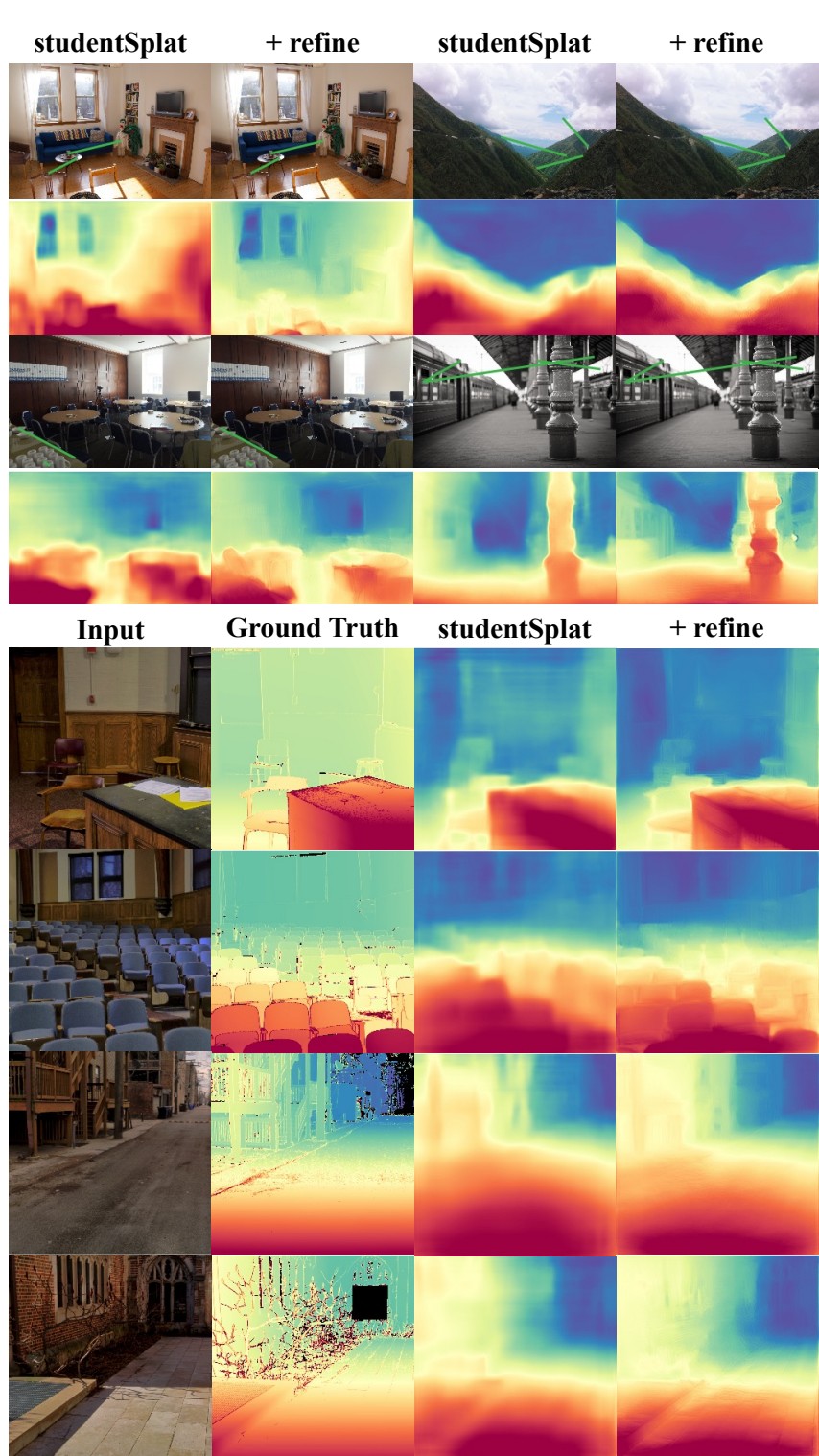

Figure 14: The qualitative comparison between the studentSplat with and without teacher refinement.

