# OpenReview forum: "studentSplat: Your Student Model Learns Single-view 3D Gaussian Splatting"
_ICLR.cc/2025/Conference — ICLR 2025 Conference Withdrawn Submission_

### Official Review · Reviewer_py2X · 2024-10-27

**Soundness:** 2
**Presentation:** 3
**Contribution:** 1
**Rating:** 3
**Confidence:** 5

**Summary:**

This paper proposes a single-view 3D scene reconstruction method. While single view reconstruction suffers from unknown scale and regions unobserved, the authors introduce two techniques. The first solution is introducing a teacher-student architecture where multi-view model acts as a teacher model to provide supervision to student model. The second solution is introducing an extrapolation network that helps to extrapolate. The method is evaluated on ACID and RE10k, and ablation studies are conducted to support author's design choices.

**Strengths:**

1. Clear writing
2. Experiments are conducted with large-scale datasets, and compared with existing SOTAs.

**Weaknesses:**

1. In page 1, line 38, why is it the first single-view 3D gaussian splatting method? I recall flash3D [1], the paper written by the same first author of the splatterimage,  has already publicly available since June 2024, which is hard to miss. I understand that the paper seems to be yet published to any conferences or journals, but this does not mean that the authors can simply ignore this already existing paper and claim author's paper as the first approach. I recommend the authors to cite this paper, as well as provide some comparisons to them. I find that their code implementations are also available.

2. From the listed contributions in Line 90~96 at page 2, none of the list points are actually this paper's contributions. For example, the first approach for single-view 3D scene GS model is achieved by Flash3D, extrapolation issue is an apparent issue that comes with single view reconstruction, which many other works also address. Finally, It is questionable whether expanding the applications of 3D Gaussian splatting model is also one of the contributions. Since there have been many attempts trying to distill the knowledge learned from multi-view images to single-view model, and the applications the authors show are not really new. Single-view depth estimation is, NVS, Text-to-3D are all already existing applications if depth/3DGS is available.

3. It is difficult for me to find technical contributions as well. Distillation from teacher to student is commonly adopted in the community and extrapolation module is simply learning from novel views, which have already been done in flash3d.

4. Finally, I believe single-view reconstruction is practically close to impossible in "estimation" tasks, where 3D geometry dominates. It is possible in "generation" tasks, but since we never know what is beyond the single view observation, I personally believe this task is ambiguous in current state. Unless the authors provide convincing arguments that justify the need of this task and the feasibility of extrapolating better than the generation model, I doubt this work will deliever a valuable message to the community.


[1] Flash3D: Feed-Forward Generalisable 3D Scene Reconstruction from a Single Image, arxiv'24

**Questions:**

Please see the weaknesses above.

---

> ### Author Response · Authors · 2024-11-12
> **Clarify the contributions and reiterate the uniqueness of our method.**
>
> Thanks for finding our presentation clear.
>
> Weakness 1:
>
> Thanks to the reviewer for pointing out the concurrent work. However, there is a fundamental difference between studentSplat and Flash3D. Flash3D requires a depth estimation model pre-trained using ground truth depth estimation which we have discussed similar methods in L126-139 and stated that they “either require[s] 3D supervision or only works at the object level” (L137-138). Technically, 3D Gaussian Splatting (3DGS) is a technique that reconstructs the 3D structure and color from multi-views and camera poses. If the 3D structures are reconstructed using ground truth annotation; only the color is reconstructed using the novel view, it should not be a 3DGS method, it is at most a method that uses 3D Gaussians as the representation. We are the first splatting method as we construct 3D Gaussians directly for multi-view supervisions instead of ground truth depth annotation. However, we recognize this ambiguity in definition and due to the presence of the concurrent work, we will modify our claim to “propose the first single-view 3D scene Gaussian splatting model that does not require relative camera poses during inference or ground truth 3D annotations.”
>
> Weakness 2:
>
> We disagree with the claim that “none of the list points are actually this paper's contributions”.
> First of all, our solution is unique and modular. In addition to extrapolating, our extrapolation design also “reduces distortion” (L92-93) as our design guide the novel-view reconstruction loss gradient from the missing context to the extrapolator but the in-context regions to the Rasterizer (L234-248). Additionally, our design automatically produces a context map (W) which allows the user to use a better extrapolation model during inference (L249-259) to fill the out-of-context region.
>
> Secondly, instead of “distill[ing] the knowledge learned from multi-view images to single-view mode”, we “bridge the gap between multi-view 3D Gaussian splatting and self-supervised single-view depth estimation” (L95) which expands the use case of 3DGS to additional tasks and potentially can enable a new way of training single-view depth estimation model. For example, current methods usually obtain depth labels from ground truth (lidar sensor), sparse labels from SfM or disparity (if multi-view data is available). Using our proposed training method (studentSplat), we have another way to obtain dense pseudo depth labels.
>
> Finally, we did not propose a new applications, rather we connected 3DGS to different applications. To make “if depth/3DGS is available” happen, unless we have ground truth 3D annotations, the current solution is to use multi-view input (with camera pose) (Flash3D uses ground truth depth label in the form of pretrained depth estimation model). We want to emphasize that studentSplat produces depth and 3D Gaussians using only multi-view training data. For text-to-3d, expanding a text-to-image method to text-to-3D is impossible for multi-view input in a feed-forward manner (MVSplat or the alike) since it is impossible for the current text-to-image method to generate two or more images with only camera view change. Although there are method for each application, given multi-view training data (no depth label), studentSplat bridges single-view 3DGS, depth estimation, and text-to-3D.
>
> Weakness 3:
>
> We agree that distillation is commonly used and we did not propose a distillation method. Our contribution is to enable single-view 3DGS using only multi-view training data; distillation is a component. Even as a componet, “extrapolation module is simply learning from novel views” is not our approach. We have commented on our extrapolation approach above in Weakness 2; our extrapolation module is designed to perform extrapolation, produce context mask, and to guide gradient flow.
>
> Weakness 4:
>
> I agree that “single-view reconstruction is practically close to impossible in "estimation" tasks” for the missing region. Therefore, our method reconstructs the 3D structure of the in-context regions (evaluated by the depth estimation Table 3). The out-of-context regions are “generated” using the extrapolator.
> In the real applications, we have a lot of 2D contents like photos and paintings. As AR and VR devices are getting more accessible, we also need 3D contents. The easiest way is to convert 2D to 3D using the proposed studentSplat. Compared with others that use GT depth information, studentSplat enjoys a larger variety of data source, lower collection cost, and wider application. Finally, “extrapolating better than the generation model” is an open problem. However, our method has the possibility to achieve better extrapolation if the field later has a better solution; our method estimates the context map (W) which allow the user to apply different extrapolation methods or generation methods during inference whereas Flash3D or the methods alike does not have this advantage.

---

> ### Comment · Reviewer_py2X · 2024-11-29
>
> I thank the reviewer for the detailed response.
> After carefully reading the response, and other reviewer's comments and the discussions between the author and the reviewers,
> I decide to keep my initial rating.

---

### Official Review · Reviewer_VAKr · 2024-11-02

**Soundness:** 3
**Presentation:** 3
**Contribution:** 3
**Rating:** 6
**Confidence:** 5

**Summary:**

The paper presents a pioneering method for single-view 3D Gaussian splatting, addressing challenges in scene reconstruction from single images. The authors introduce a teacher-student architecture, where a multi-view teacher model provides geometric supervision, mitigating scale ambiguity. Additionally, an extrapolation network enhances scene context completion, leading to high-quality reconstructions. The proposed method achieves state-of-the-art results in single-view novel-view reconstruction and demonstrates potential in self-supervised depth estimation, broadening the applicability of 3D Gaussian splatting models​.

**Strengths:**

* The approach of leveraging geometric priors from multi-view reconstruction methods to enhance single-view reconstruction is intriguing, and the authors have experimentally demonstrated significant improvements in the perspective extrapolator through multi-view distillation.
* The authors present a simple yet effective method for extrapolating when computing the novel view reconstruction loss.
* The paper is well-written and easy to follow.

**Weaknesses:**

* **3D Consistency**: In the unseen regions when extrapolating new views, the rendered results depend on the 2D generative model MI-GAN. Therefore, I am skeptical about the model's ability to generate extrapolated continuous new views with 3D consistency. I recommend that the authors supplement the discussion with relevant visual results or theoretical analyses.
* **Overclaim**: The authors state in the contributions section that they "propose the first single-view 3D scene Gaussian splatting model that does not require relative camera poses during inference." However, to my knowledge, there exists a single-view 3DGS-based method, Flash3d [1], which can achieve much of what this work does without needing camera poses. I recommend that the authors explicitly address how their method compares to Flash3D and explore the differences in their approach. Additionally, if feasible, including comparative experiments in their analysis would enhance the robustness of their claims.

[1] Szymanowicz S, Insafutdinov E, Zheng C, et al. Flash3D: Feed-Forward Generalisable 3D Scene Reconstruction from a Single Image[J]. arXiv preprint arXiv:2406.04343, 2024.

**Questions:**

* Given the model's relatively low parameter count and the use of a lightweight generative model, does this method offer any advantages in terms of training time and inference speed compared to other single-view methods?
* Is the generative model MI-GAN fixed or fine-tuned during training? Could the authors provide more detailed experimental settings and analyses regarding MI-GAN?
* The authors propose a method for refining student output in Appendix D. Can this method be utilized during the training process of the student or teacher models as a form of self-supervised approach to enhance multi-view consistency?

---

> ### Author Response · Authors · 2024-11-12
> **Clarification on the difference between tasks, extrapolation, and our contributions. Answer the questions.**
>
> Thanks for the comments on the effectiveness of our method.
>
> First of all, we want to clarify the difference between tasks.
>
> Novel-view reconstruction: aims to reconstruct the novel-view instead of 3D structure.
>
> Depth estimation: aims to estimate the 3D depth instead of novel-view.
>
>
> Weakness 1:
>
> We agree with the review that the 3D consistency in the extrapolated region is not assured. However, we never aimed to reconstruct the 3D structure of the extrapolated region. Similar, the teacher model (MVSplat) also did not assure the 3D consistency in the missing region. As stated in L226, the purpose of the extrapolator is to improve the novel-view reconstruction instead of performing 3D structure reconstruction in the missing region. Note that in the novel-view reconstruction task, we do not require reconstruction of 3D structure.
>
> The “3D Consistency” is ensured in the in-context region and we demonstrated it through self-supervised single-view depth estimation task (Table 3).
>
>
> Weakness 2:
>
> Thanks to the reviewer for pointing out the concurrent work. We agree that including it will improve the overall presentation of our paper. However, there is a fundamental difference between studentSplat and Flash3D. Flash3D requires a depth estimation model pre-trained using ground truth depth estimation which we have discussed similar methods in L126-139 and stated that they “either require[s] 3D supervision or only works at the object level” (L137-138). In the application setting, this difference will rule out Flash3D as a self-supervised depth estimation method. Technically, 3D Gaussian Splatting (3DGS) is a technique that reconstructs the 3D structure and color from multi-views and camera poses. If the 3D structures are reconstructed using ground truth annotation; only the color is reconstructed using the novel view, it should not be a 3DGS method, it is at most a method that uses 3D Gaussians as the representation. However, we recognize this ambiguity in the defination and due to the presence of the concurrent work, we will modify our claim to “propose the first single-view 3D scene Gaussian splatting model that does not require relative camera poses during inference or ground truth 3D annotations.”
>
> Question 1:
>
> It is about 2 times faster to train and test than splatterImage. However, we are proposing the training method and different models can be used directly with our proposed training method. Therefore, we simply used a popular model design.
>
> Question 2:
>
> MI-GAN was fine-tuned during training. We selected MI-GAN only because of its efficiency. As discussed in L249-259, we do not aim to produce perfect extrapolation results. As long as we can generate a mask (W) of where the in-context and out-of-context regions are, we can always use a more powerful extrapolation model during inference.
>
> Question 3:
>
> The proposed method for refining student output uses the multi-view teacher model to perform 3DGS on the output of the student model along with the single-view input image. Basically, we aim to also produce 3D Gaussians for the extrapolating region since the extrapolator only extrapolates in the 2D reconstructed view. As suggested by the reviewer, the teacher model can also be added after the student model during training to improve the performance while still requiring only one image during inference. However, it will not add too much to our existing training pipeline and should not provide additional insight compared to what we have in Appendix D.

---

### Official Review · Reviewer_B9uo · 2024-11-02

**Soundness:** 3
**Presentation:** 3
**Contribution:** 2
**Rating:** 5
**Confidence:** 4

**Summary:**

This paper introduces studentsplat. Inspired by some recent feed-forward 3D Gaussian Splatting methods with multi-view or single-view inputs, studentsplat combines two settings and uses multi-view model (teacher) to improve the performance of single-view model (student). An extrapolation network (GAN) is used to complete missing scene context and thus facilitates training. The idea is straightforward and the performance is better than recent methods in single image rendering and monocular depth quality.

**Strengths:**

The idea of distilling knowledge from multi-view model to single-view model is simple and makes sense.

The paper is clearly written.

Studentsplat outperforms recent methods in single image rendering and monocular depth.

The ablation study in main paper and supplementary is thorough.

**Weaknesses:**

I felt the evaluation is not very convincing. In single image setting, it is not surprising that pixelsplat and mvsplat perform worse since they rely on feature matching across different views, which is unavailable with single image (or let’s say two same images with baseline=0). For the evaluation with single image, my concerns mainly come from Fig. 3, 4, 7, 8. In these images, we can find that the viewpoints of target views and input context view are similar (i.e. the baseline between input view and target view is small). In some cases, I think directly copying the input image as output would provide nice rendering metrics as well. Therefore, I think evaluation with large baseline between input context view and target view should be included.

In Fig. 11, seems the rendered depth from studentsplat is usually oversmoothed. Additionally, since the authors also mentioned Depth Anything (the student model also has similar structure as Depth Anything, i.e. DINOv2 + DPT), comparison with Depth Anything on depth quality can be included.

**Questions:**

L90: The claim ‘Propose the first single-view 3D scene Gaussian splatting model’ is not convincing. Though SplatterImage (CVPR 2024) mainly focused on object-level scenes in the paper, the results in Table. 1 show that SplatterImage perform relatively well on large scenes (a little worse than the proposed method).

Can you show some visualization of extrapolation with MI-GAN?

L1009-1015: The visualization looks wrong. The input images and target image are from different scenes.

Typo: L237: compositing-> composition

---

> ### Author Response · Authors · 2024-11-12
> **Clarify the contributions and answer the questions**
>
> Thanks for the comments about our clear presentation and strong model performance.
>
> Weakness 1:
>
> First of all, we agree that the context and target are not far enough. We use the same views as proposed by previous works. Making the context view and target view more different will likely reduce the performance for all the methods evaluated but not affect the ranking of the methods. Thus, we followed previous works instead of proposing our own evaluation.  Additionally, the gap between the best performing model and the worst performing model (Table 1) is large enough to suggest that the evaluation is valid and simply copying the input as the target will produce bad results.
>
> Secondly, to make sure there is 3D geometric structure is learned, the best way is to evaluate the geometric structure directly. Therefore, we evaluated the single-view depth estimation performance in Table 3 in addition to the novel-view reconstruction performance. Note that previous works are only evaluated on novel-view reconstruction.
>
>
> Weakness 2:
>
> Note that our model in the single-view depth estimation setting is purely self-supervised despite having the same architecture as Depth Anything. Therefore, comparing it against a large-scale supervised method is unfair. Additionally, we are not restricting the architecture. We only proposed the training method, the model architecture can be changed. Finally, being smoother compare to the ground truth is again due to the lack of ground truth supervision. But compare to the other self-supervised methods (GasMono, SpaltterImage, Figures 9 and 10), studentSplat is reasonably sharp.
>
>
> Question 1:
>
> First of all, we disagree with the author that the performance improvement over SplatterImage is “a little”, the improvements are at least 5% for all datasets. For novel-view reconstruction, studentSplat not only outperforms the splatterImage on the quantitative results (Table 1), it also has much less distortion (Figure 3, Appendix Figure 8). For single-view depth estimation, studentSplat is comparable to a self-supervised single-view depth estimation method (GasMono) while splatterImage is much worse (Table 3). Additionally, splatterImage has a lot of noise whereas studentSplat is much clearer (Appendix Figure 9).
>
> Question 2:
>
> Figure 5 shows the difference between the extrapolator and w/o extrapolator and appendix Figure 7 shows the weight for the original context and the extrapolating target (which indicating the extrapolating region).
>
> Question 3:
>
> Thanks for noticing the visualization error, we will correct it.
>
> Question 4:
>
> We will correct typos.

---

> ### Comment · Reviewer_B9uo · 2024-11-27
>
> First, other reviewers mentioned Flash3D (it is on arxiv in early June, 4 months before ICLR deadline). I think it should be discussed. Additionally, the claim throughout the paper that the paper ‘propose the first single-view 3D scene Gaussian splatting model’ is more unconvincing. In authors' response to reviewer, it is said the claim will be modified as “propose the first single-view 3D scene Gaussian splatting model that does not require relative camera poses during inference or ground truth 3D annotations.” However, this description is still not accurate. Flash3D uses pretrained MDE model and does not use depth supervision during its training process (rendering loss only).
>
> Second, for my question about visualization of MI-GAN, I misunderstood during my initial review and though MI-GAN is only used during training. sorry about this.
>
> Third, I am not satisfied with the authors' response about large-baseline comparison. The authors can simply choose some test pairs with large-baseline and then compare with other methods. Will it be difficult?
>
> Fourth, as pointed-out by Reviewer Sorz, it makes sense to use pretrained MDE since the proposed method deals with very challenging single-view reconstruction (rendering and depth estimation). About comparison with depth anything, the authors already had one ablation model with pretrained weight of depth-anything in Table 6. Thus I believe authors can compare this ablation model with pretrained depth anything on depth quality.
>
> In summary, I am not convinced to improve my rating to accept this paper.

---

### Official Review · Reviewer_Sorz · 2024-11-04

**Soundness:** 3
**Presentation:** 3
**Contribution:** 2
**Rating:** 3
**Confidence:** 4

**Summary:**

The paper introduces studentSplat, a single-view 3D Gaussian splatting (3DGS) aimed at scene-level reconstruction from a single image. Recognizing challenges in single-view reconstruction, such as scale ambiguity and context extrapolation, the authors propose a teacher-student architecture. Here, a multi-view teacher (pre-trained feed-forward Gaussian Splatting Model (e.g., MVSplat)) provides geometric supervision to a single-view student model during training, addressing scale ambiguity and encouraging geometrically valid reconstructions. Additionally, the model uses an extrapolation network to adaptively fill missing scene context through GANs, improving the novel-view reconstruction quality.

**Strengths:**

- The overall paper is well-written with the architectural designs mentioned in detail making the readers easy to understand the training procedure and contributions of the work.
- Although the task being challenging, the proposed method shows strong performance, achieving state-of-the-art in multiple datasets.
- The proposed method is efficient in terms of the model parameters and the number of Gaussians compared to previous methods.

**Weaknesses:**

- Mitigating the use of camera poses : The authors mention that the camera pose of a single image can be defined as the identity matrix, mitigating the use of camera poses of multi-view images. However, during the teacher-student geometric supervision, as MVSplat[1] has been trained on both RealEstate10K and ACID using SfM Camera poses, this supervision guides studentSplat to learn this SfM Camera Pose scales which enables the photometric loss of $L_{photo}$ with a specific relative camera pose $[R|t]$. As a result, I strongly believe that the current training scheme cannot be claimed as mitigating the camera pose.
- Performance contribution of extrapolator : As the evaluation is currently done with 2 novel views outside and one novel view inside the context frustums, I agree that the extrapolation performance of StudentSplat is superior than other methods and the teacher MVSplat. However there is no qualitative or quantitative results of the performance of studentSplat without the extrapolator which makes it hard to fairly compare with other methods.

**Questions:**

- As mentioned in the weakness section, training the student network in the same dataset with the teacher network cannot be claimed as fully mitigating the camera pose constraint. Can the model be trained in a new dataset the teacher has not been trained on? Or can the student network be trained with the rendered results of the teacher instead of ground truth images?
- To fully understand the performance of StudentSplat without the extrapolator, can the authors provide qualitative and quantitative results without the extrapolator?
- Extended from the previous question, I am concerned that the performance of studentSplat without the extrapolator is similar to the performance of MVSplat with Gaussians from one context images. As a result, can the authors show the performance of MVSplat (one view) + extrapolator?

**Details Of Ethics Concerns:**

There are no concerns.

---

> ### Author Response · Authors · 2024-11-12
> **Clarify what the reviewer has missed from our paper and answer the questions.**
>
> Thank you for you comments about the strong performance and efficiency of our studentSplat.
>
> Questions 1:
>
> As stated in our contribution our method “does not require relative camera poses during inference” (line 90). We will still require camera poses (SfM) during training. It is impossible to train a novel-view reconstruction method using only an image dataset. Therefore, like all other novel-view reconstruction methods, our model needs to be trained on a multi-view dataset (either video or multi-view). We then need to perform SfM to obtain the camera poses for the training data. However, we do not need relative camera poses during inference as claimed. During inference, only one “fake” camera pose (any camera pose defined by the user) is needed for the entire testing dataset since we only need one input image. Therefore, no real camera pose is needed. To further support the claim of “mitigating the camera pose”, we performed the single-view depth estimation using the one input image and the same camera view (for the entire dataset) defined by the user (Table 3).
>
>
> Question 2:
>
> There are both qualitative (Figure 5) and quantitative results (Table 4) of the performance of studentSplat w/o extrapolation in the ablation study.
>
>
> Question 3:
>
> First of all, in Table 4, we can already see that the performance of studentSplat without extrapolator (PSNR 21.38, SSIM 0.741, LPIPS 0.208) is significantly better than MVSplat (PSNR 17.73, SSIM 0.585, LPIPS 0.296). Secondly, MVSplat (one view) + extrapolator will require training as the extrapolator cannot work with a 3DGS method out-of-the-box. As studentSplat without extrapolator already outperform MVSplat by a large margin, adding extrapolation should not flip the result.

---

> ### Comment · Reviewer_Sorz · 2024-11-19
> **Response to Author**
>
> First of all, thank you for clarifying the points I have missed in my initial review. However, I have a few concerns and questions that remain. I would appreciate a further discussion of these points.
>
> To start with my initial questions,
> For Q1, as the authors mentioned, it is true that studentSplat does not require any camera pose during inference and my concerns are resolved. However, I find that this is not the "first" attempt as Flash3D[1] has also tackled a similar problem. Therefore, I believe that the presentation should be toned down a bit.
>
> One additional question is, if we still need camera poses during training, what is the fundamental advantage of utilizing such a Teacher-Student framework, especially using MVsplat as a teacher? Similar to Dust3r[2] and its utilization in InstantSplat[3], I belive utilizing the ground truth depth information as supervision is much more advantageous than utilizing MVSplat as two-view geometry is also extremely difficult to provide accurate geometries. In addition, as a feed-forward framework, MVSplat also can only work well in its trained domain, which is much smaller than existing depth datasets.
>
> For Q2 and Q3, I observed the results in Table 4 and Figure 5 in my initial review, but I did not understand where the improvements of studentSplat mainly comes from. My concerns still remain that the improvements are from the adaptation of the extrapolation network and the architectural design that can take single images as input. Although studentSplat shows improvements over MVsplat without the extrapolation network, as MVSplat was not trained to take the same images with identical poses as inputs, I believe that the performance of MVSplat is largely degraded due to this input domain gap from its training and evaluation. However, I agree with the authors that there is no direct way to compare with MVSplat and request a direct qualitative comparison with MVsplat and studentSplat w/o extrapolator given the same image.
>
> After the discussion and reviewing the paper again, I have some additional questions.
> 1. **The importance of the task**: I am not fully convinced why we need a single-view 3D Gaussian Splatting model. When provided a single image it is impossible to estimate the accurate geometry of the scene without any priors or additional information such as additional images with the relative camera pose or ground-truth depth. Therefore, I believe this task should be solved with additional priors such as training with large ground-truth depth datasets, generative priors similar to state-of-the-art Monocular Depth Estimation Networks such as DepthAnythingv2[4], Marigold[5], MOGE[6].
>
> 2. **Randomness in the Extrapolator**: As studentSplat takes single images as input, I believe its application should be in extrapolation. However, I believe that there are no architectural designs that regularize the extrapolated regions to be consistent throughout the estimation. This largely constraints the network's downstream application as only up to two views can be consistent. Given such limitations, I do not find a significant advantage of using studentSplat.
>
> I believe that these two additional concerns make the contribution and application of studentSplat extremely weak. For, single-view depth estimation, it is much easier to utilize pre-trained MDE networks[4,5,6]. For Novel View Reconstruction & 3DGS, the network can only reconstruct consistent geometry up to two views due to the randomness of the extrapolator. For text-to-3D, I believe that adding a small optimization phase of Gaussian parameters with the position estimated by MDE will produce a better estimation of geometry.
>
> As a result, with my current concerns, I believe that this work is not enough for ICLR in the current phase. However, with my concerns properly resolved I am eager to raise my scores towards acceptance.
>
> ---
> [1] Szymanowicz, Stanislaw, et al. "Flash3D: Feed-Forward Generalisable 3D Scene Reconstruction from a Single Image." arXiv preprint arXiv:2406.04343 (2024).
>
> [2] Wang, Shuzhe, et al. "Dust3r: Geometric 3d vision made easy." Proceedings of the IEEE/CVF Conference on Computer Vision and Pattern Recognition. 2024.
>
> [3] Fan, Zhiwen, et al. "Instantsplat: Unbounded sparse-view pose-free gaussian splatting in 40 seconds." arXiv preprint arXiv:2403.20309 2 (2024).
>
> [4] Yang, Lihe, et al. "Depth Anything V2." arXiv preprint arXiv:2406.09414 (2024).
>
> [5] Ke, Bingxin, et al. "Repurposing diffusion-based image generators for monocular depth estimation." Proceedings of the IEEE/CVF Conference on Computer Vision and Pattern Recognition. 2024.
>
> [6] Wang, Ruicheng, et al. "MoGe: Unlocking Accurate Monocular Geometry Estimation for Open-Domain Images with Optimal Training Supervision." arXiv preprint arXiv:2410.19115 (2024).

---

> > ### Author Response · Authors · 2024-11-19
> > **Clarify the difference between our approach and answer reviewer questions**
> >
> > Thanks for the thoughtful comments and glad we address some of the reviewer's concerns.
> >
> > Before getting into the additional concerns, we would like to clarify the difference between studentSplat and methods like Flash3D. studentSplat does not require ground truth depth label whereas method like Flash3D requires a model trained using ground truth (GT) depth label (or GT model). 3D Gaussian Splatting (3DGS) is designed to reconstruct the 3D structure of a scene given only different camera views (images+poses). Therefore, using ground truth annotation as part of the training is technically not a 3DGS method, it (Flash3D or the alike) uses 3DGS as a part of the method to perform 3D reconstruction. However, we agree there may be some ambiguity in the definition and we will modify the claim to "propose the first single-view 3D scene Gaussian splatting model that does not require relative camera poses during inference or GT 3D annotations."
> >
> >
> > Question 1:
> >
> > As the reviewer suggests, one would need either GT 3D labels or multi-view supervision to reconstruct 3D from 2D. We want to emphasize that studentSplat follows "multi-view supervision" approach thus it can correctly estimate the 3D structure (as shown in the depth estimation performance in Tab. 3); we reconstruct 3D geometry using multiple views (thru a multi-view teacher) and reconstruct the 3D photometric parameters using the novel-view supervision. If we only consider 3D geometry (without photometric parameters), our method can be reduced to performing single view depth estimation using pseudo 3D labels from a multi-view model. As the result, our method is also a self-supervised depth estimation method (Tab. 3) and achieves geometry reconstruction. This is one of the advantage of our method: while other methods (e.g., Flash3D) requires GT 3D to reconstruct novel-views, our method not only illuminate this requirements, it also performs self-supervised single-view 3D (depth) estimation which the supervised method (Flash3D) cannot perform.
> >
> > The reasons why we need a self-supervised single-view 3DGS method like studentSplat are twofold.
> >
> > First, as a self-supervised method, our method scale better compare to the supervised method (Depth Anything, Flash3D etc). Since video data are ubiquities, we can collect a extremely large and diverse dataset and extract the camera pose using structure-from-motion (images+poses are all we need for training). If we were to train a supervised method, we have a very limited data sources with 3D label (e.g., Game engine or lidar data). Because of this self-supervised nature, studentSplat has a larger potential especially when the application domains do not have 3D label.
> >
> > Second, as VR device become more accessible, there needs an efficient way for 3D content creation. As the 2D contents are ubiquities (e.g., images on a smartphone), we can easily populate 3D content by running studentSplat on 2D content.
> >
> > Question 2:
> >
> > As suggested in our response to Q1, the application is not only in extrapolation; our method can be used as a depth estimation method which does not require extrapolation. Even when applying in the extrapolation setting, we can achieve more than 2 consistent views. Before we explain how to achieve consistent views using our method, we would like to first emphasize the extrapolator design.
> >
> > Our extrapolator (Line 233-259) produces a context map W in addition to extrapolation, which allows us to identify the in-context region and the out-of-context region. This will allows for a better extrapolation model to be used to improve the out-of-context region while maintain the accuracy of the in-context region.
> >
> > Next, to construct more than two consistent views, we refer to Appendix D (line 1242-1269). We summarize some steps below:
> >
> > 1. From one input image v_1 and a user defined starting camera pose p_1 (can be random), we first construct the 3DGS.
> >
> > 2. Then, we move p_1 to render at p_2 (user defined). Next, we perform extrapolation and obtain view v_2 and W_2. Note that when we are satisfied with v_2, we can perform next step. Otherwise, we can use a better extrapolator and W_2 to get a better extrapolation.
> >
> > 3. Note we now have two consistent views. We know that a multi-view method will reconstruct more than 2 consistent views thus we only need to feed (v_1,p_1) and (v_2,p_2) to the multi-view teacher to generate the rest of consistent views. If we again need to extrapolate, we feed the output v_3 from the teacher to the extrapolator to get W_3 and use a better extrapolation if needed. Because of the existence of W_3, we know what to not modify the already generated consistent views.
> >
> > Note that the above method achieves better geometric reconstruction (depth estimation Tab. 8 line 1272-1290 and Fig.14 Pg 25) although we did not report it in the main result.
> >
> >
> >
> > Finally, we did not aim for the best performance in all applications; we intend to provide an alternative, unified, self-supervised, and scalable approach.

---

> > > ### Comment · Reviewer_Sorz · 2024-11-20
> > > **Response to author**
> > >
> > > After reading through the responses from the authors, none of my concerns have been addressed.
> > >
> > > To start with the importance of the task,
> > >
> > > **I am not convinced with the self-supervised approach:** It is a common approach in depth estimation networks to undergo self-supervised approaches due to the difficulty of obtaining 3D annotations. However, using readily available Monocular Depth Estimation networks trained with these existing datasets is not a big bottleneck. Although the authors argue that Flash3D requires 3D annotations, they simply use a pre-trained MDE network without further training. This is definitely not a big bottleneck. Furthermore, if the authors are adopting a self-supervised strategy due to scalability, there should be enough evidence such as:
> > > 1. The proposed method is scalable: Other than the fact the training can simply be scaled-up, there should be evidence that by providing more training data, the model is indeed scalable.
> > > 2. Scalable than pre-trained MDE networks: As the authors argued that the proposed method could be scaled further than MDE networks that require 3D annotations, the method should be more scalable than MDE networks. However, the teacher network MVSplat is trained either only on the RealEstate10K dataset or the ACID dataset. These datasets are much smaller than compared to the scale MDE networks are trained on, and as studentSplat largely depends on the teacher network, I believe that scalability is not a contribution of this work.
> > >
> > > **Lack of Contribution**: It would have been more interesting if the proposed method adopted a teacher network that did not require any additional data such as camera pose or if the teacher network is generalizable to any existing in-the-wild images. However, MVSplat requires an accurate camera pose for high-quality estimation and is not generalizable to in-the-wild images that are out-of-distribution of the training data. I am not convinced of simply adopting MVSplat as a teacher network.
> > >
> > > **Application Strategy**: Although the authors argue that the construction of more than two views can be possible, it largely depends on an external method MVSplat which is not the proposed work's contribution. In addition, even if we take the approach of single-view extrapolation with studentSplat -> MVSplat -> studentSplat iteration framework, as studentSplat cannot take more than one image if we assume the scenario that the camera trajectory goes to the right and comes back to the left again, the extrapolated region from studentSplat will also be inconsistent which will cause MVSplat to fail. As a result, I am not convinced why such a single-view reconstruction method can be applied in a practical scenario, other than a artificial scenario.
> > >
> > > With my concerns remaining, I will keep my initial rating (reject).

---

> > > > ### Author Response · Authors · 2024-11-20
> > > > **Addition comments on the misunderstanding**
> > > >
> > > > Thanks for the quick response. The reviewer has valid points but seems to have some misunderstanding on my previous comments. Because we were asked to explain the importance of the task, we try to explain the potential more. In this response, we aim to bring the attention back to our contribution.
> > > >
> > > > **Self-supervised approach and scalability:**
> > > > Even without out scalability, adaptation of self-supervised strategy connects depth estimation with 3D gaussian splatting by training one model that perform both task. Moreover, we our main argument is that single-view 3DGS is possible without additional supervision. In this sense, the existence of method that uses MDE model is irrelevant to our contributions. We are aware the availability of a pre-trained MDE model will improve the performance (Table 6 in the Appendix.) but our goal is to provide an alternative approach and to show a possibility to achieve single-view 3DGS. We have shown the potential for studentSplat to also serve as a self-supervised depth estimation method by comparing to another self-supervised depth estimation method (GasMono, one of the common self-supervised approach referred by the reviewer).
> > > >
> > > > As for the scalability, 1. Our contribution is enabling single-view 3DGS while still maintain the self-supervised nature of 3DGS; our previous explanation is all about the scalability of the data collection (both RealEstate10K and ACID are unlabeled and the camera poses are obtained using Structure-from-motion). By using the scalable data source, our method can potentially be more scalable. 2. The teacher model can be further trained using the larger data as well or can be replaced completely. If we were not to maintain the self-supervised nature, even a MDE model can be used as the teacher model. Therefore, what the current teacher model is trained on should not effect our contribution.
> > > >
> > > > Finally, commenting on the importance of the task and the use of MDE. A multi-view method can also benefit from pre-trained MDE model but they intentionally choose to not do so as application of MDE is an obvious answer which does not provide too much insights. Our approach shares the same motivations but we further expand the applications.
> > > >
> > > >
> > > > **Lack of Contribution**: We want to reiterate that out contribution comes from connecting self-supervised depth estimation and 3D gaussian splatting and providing a method single-view 3DGS method without ground truth annotation. Therefore, which teacher model to adopt should not effect our contribution as long as the teacher model is also self-supervised. Adopting MVSplat is an implementation choice for efficiency; any multi-view method can be used. Additionally, "MVSplat is not generalizable to ...  out-of-distribution of the training data" is false as MVSplat and our studentSplat both demonstrated cross-dataset generalization (Table 2).
> > > >
> > > > **Application Strategy**: First, our contribution in this part is to use a multi-view model in addition to a single-view model to mitigate the need of camera poses. We only require a multi-view model not a specific model. Additionally, the statement that "studentSplat -> MVSplat -> studentSplat" cannot do deal with the scenario that " the camera trajectory goes to the right and comes back to the left again" is false. In fact, there is a misunderstanding of the iterative pipeline; there real pipeline can be "studentSplat -> MVSplat -> extrapolator -> MVSplat". Once we have the initial view generated, we will not need to use studentSplat anymore as we now have multi-view input; we only need to apply the extrapolator to complete the missing region. More importantly, even if we assume two views with minor inconsistency, since multi-view method produces only one 3D Gaussian representation, the reconstructed views will still be consistent although the quality will be slightly lower (will not cause MVSplat to fail due to inherited adaptability of neural network).
> > > >
> > > > More importantly, to comment on the randomness of extrapolator, extrapolation is still an open question in 3DGS and not many work is tackling this issue; even the multi-view methods cannot perform extrapolation by themselves. Even if consistency is an potential issue, it should not reduce effect our other contributions: providing a context map to enable the application of better extrapolation model and converting single-view 3DGS to multi-view 3DGS.

---

> ### Comment · Reviewer_Sorz · 2024-11-28
> **Response to author**
>
> First of all, I would appreciate if the authors could re-clarify the main contribution of their work. From the discussions so far, I understood that the main contribution was the self-supervised formulation of single-view 3DGS from the claim from the authors like "propose the first single-view 3D scene Gaussian splatting model that does not require relative camera poses during inference or GT 3D annotations." and the keywords  "self-supervised learning" for this work.
>
> Therefore, I believe claims such as "If we were not to maintain the self-supervised nature, even a MDE model can be used as the teacher model. Therefore, what the current teacher model is trained on should not effect our contribution." **cannot be made** for this work.
>
> By regarding "self-supervised training" as the main contribution of this work, my question and concern is "What is the advantage of self-supervised training for single-view 3DGS?". However, the answers and discussions with the authors did not effectively resolve my concerns. I have explained below why my concerns are left unresolved:
>
> Although the authors differ in self-supervised depth estimation and 3DGS learning, I believe the two are highly related tasks. Also mentioned in this work, estimating accurate geometry from the images (positions of the Gaussians) is the main challenge of learning Gaussian parameters. Therefore, I strongly believe that the authors should mainly compare the effectiveness, and practicality over utilizing pre-trained MDE networks.
>
> Regarding scalability, the authors are simply assuming that due to the cross-generalizability of MVSplat from Re10K -> DTU, the MVSplat model can be scaled to more data when compared to pre-trained MDEs. This assumption is incorrect, more specifically, the PSNR scores in DTU which are 9, 12, 14dB rather verify that these models are not enough to be called a fully generalizable approach. PSNR 14 can be simply achieved with mean colors due to its ambiguity in calculation and 9 of MVSplat shows that its model lacks generalizability. Although the authors claim that studentSplat can be scaled with a simple pair of images with the pose, as SfM algorithms require a large number of images to estimate accurate poses, more than a pair of images are required to obtain the pose. In addition, even the images with pose, studentSplat depends on the accuracy of MVSplat which is less generalizable than MDE networks.
>
> From my concerns being unresolved, I will keep my score. I also want to request the authors that given the opportunity to refine the original paper and discuss it with the reviewers for 3 weeks, I encourage the authors to show results from additional experiments.

---

### Note · Authors · 2024-12-05

**Comment:**

We thank all the reviewers for their time and effort. Due to the extended scope of the reviewers commons and constrains on additional experiments, we decide to withdraw this submission. Wish all the reviewers the best luck in their submissions.

**Withdrawal Confirmation:**

I have read and agree with the venue's withdrawal policy on behalf of myself and my co-authors.